# Multi-objective Large Language Model Alignment with Hierarchical Experts

**Zhuo Li**[1*]**, Guodong Du**[1*]**, Weiyang Guo**[1]**, Yigeng Zhou**[1]**, Xiucheng Li**[1]**,**
**Wenya Wang**[2]**, Fangming Liu**[3,4]**, Yequan Wang**[5,6]**, Deheng Ye**[7]**, Min Zhang**[1]**, Jing Li**[1✉]
[1]Harbin Institute of Technology, Shenzhen, China
[2]Nanyang Technological University, Singapore    [3]Peng Cheng Laboratory, China
[4]Huazhong University of Science and Technology, China
[5]Beijing Academy of Artificial Intelligence    [6]Peking University    [7]Tencent
zuoer190191@mail.ustc.edu.cn    jingli.phd@hotmail.com

## Abstract

Aligning large language models (LLMs) to simultaneously satisfy multiple objectives remains a significant challenge, especially given the diverse and often conflicting nature of human preferences. Existing alignment methods struggle to balance trade-offs effectively, often requiring costly retraining or yielding suboptimal results across the Pareto frontier of preferences. In this paper, we introduce HoE (Hierarchical Mixture-of-Experts), a *lightweight*, *parameter-efficient*, and *plug-and-play* approach that eliminates the need for model training, while enabling LLMs to adapt across the entire Pareto frontier and accommodate diverse user preferences. In particular, HoE consists of three hierarchical components: LoRA Experts, Router Experts and Preference Routing, reaching state-of-the-art Pareto frontiers and achieving a trade-off between parameter size, training cost, and performance. We evaluate HoE across various tasks on 16 objectives and 200 different preferences among 8 benchmarks, demonstrating superior performance over 15 recent baselines. WARNING: This paper contains potentially offensive text.

## 1 Introduction

Large language models (LLMs) have achieved remarkable success in aligning with broadly defined human values (Achiam et al., 2023; Ziegler et al., 2019; Sun et al., 2023). However, human preferences in practice are highly diverse and cannot be fully captured by an universal alignment goal. Users may pursue multiple personalized objectives, and even when the objectives are the same, their relative importance often varies across individuals and contexts (Fu et al., 2024; Yang et al., 2024c; Lin et al., 2025; Ren et al., 2025; Zhang et al., 2025). Existing approaches (Sun et al., 2023; Yang et al., 2025; Chen et al., 2025), which typically optimize for one objective or a fixed combination, fall short in flexibly covering this preference space. These observations highlight the necessity of multi-objective alignment (MOA) to enable scalable and preference-steerable LLMs (Lin et al., 2025; Guo et al., 2024; Ramé et al., 2023; Jang et al., 2023; Yang et al., 2024c; Lin et al., 2025; Chen et al., 2025).

The central difficulty of multi-objective alignment (MOA) lies in its inherently steerable nature: LLMs must dynamically adapt to arbitrary user preferences rather than a single fixed goal, essentially acting as a "jack-of-all-trades" (Lin et al., 2025; Chen et al., 2025; Du et al., 2025a). This steerability introduces two major challenges. *First, objectives often conflict with each other.* Parameters tuned to improve one objective (e.g., helpfulness) often undermine another (e.g., harmlessness) (Chen & Kwok, 2025; Zheng & Wang, 2024; Du et al., 2024; Yadav et al., 2023; Zhou et al., 2024a). *Second, competition also even exists across preferences along the Pareto frontier* (Wang et al., 2024b; Shi et al., 2024; Zhou et al., 2024b; Li et al., 2021). For instance, a model trained uniformly across all weightings (black solid Pareto Frontier in Fig. 1 Left) cannot achieve optimal performance at a specific weighting (e.g., [0.5, 0.5]), compared to the expert model fine-tuned exclusively for that preference (colored dashed Pareto frontier in Fig. 1 Left).

---

✉ Corresponding author. * Equal contribution. The code is publicly available at: this https URL.

To overcome this limitation, *we adopt a decomposition-based strategy for MOA, breaking down the multi-objective alignment problem into a series of single-preference subproblems (Zhang & Li, 2007), each associated with a set of specialized parameters.* These parameters, referred to as experts, are each assigned to a distinct preference, focus solely on their corresponding preferences and optimize within their localized subproblem regions. This strategy avoids the pitfalls of a single monolithic model attempting to cover the entire Pareto Frontier, thereby circumventing the steerability bottleneck.

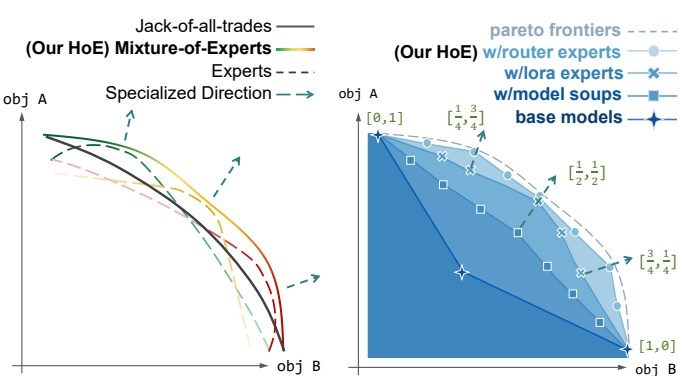

Figure 1: (Left) HoE decomposes the multi-objective alignment problem into a series of single-preference subproblems, each handled by a specialized expert. (Right) HoE employs hierarchical experts, integrating LoRA and router experts to approach near-optimal Pareto frontier.

Building on this idea, we then instantiates each expert as a lightweight LoRA adapter within a Mixture-of-Experts framework. Preference-specific behavior is captured in small, composable LoRA modules while the backbone parameters remain shared. At inference time, a preference-conditioned routing mechanism composes and activates the appropriate LoRA experts, adapting the model's behavior dynamically to realize arbitrary weightings. By this design, we efficiently reconstruct the full Pareto frontier from a collection of localized, preference-specific experts, enabling a scalable, steerable, and parameter-efficient approach for multi-objective alignment.

In this work, we propose HoE, a novel hierarchical Mixture-of-Experts framework for multi-objective alignment. HoE is a ***lightweight***, ***parameter-efficient***, and ***plug-and-play*** solution that eliminates the need for training any models while achieves strong performance across the entire Pareto frontier. It combines the decomposition principle with the LoRA-based MoE design to enable scalable, efficient and fine-grained control over the entire Pareto frontier. Specifically, HoE comprises three hierarchical components: LoRA experts, router experts, and preference routing. (1) "LoRA experts" are first extracted *without training* from off-the-shelf single-objective models using task-vector singular value decomposition (task-SVD), capturing distinct alignment objectives in compact adapter modules. (2) "multi-objective LoRA experts" are then synthesized, also *without training* by merging multiple existing single-objective LoRA experts, to enable on-demand generation of alignment capabilities across arbitrary preference configurations. (3) "Router experts" are trained with negligible parameters to dynamically select and combine the appropriate experts based on user-specified preferences, allowing efficient traversal of the Pareto frontier. Through this hierarchical design, HoE not only provides precise control over preference-specific behavior but also balances alignment performance, parameter cost, and training efficiency. It serves as a practical and effective solution for scalable multi-objective alignment in LLMs.

The main contributions of this study are as follows:

- We investigate a novel decomposition strategy that breaks down the multi-objective alignment problem into a series of single-preference subproblems, each handled by a set of specialized experts, enabling fine-grained control and full Pareto coverage.

- We propose HoE, a *lightweight*, *parameter-efficient* and *plug-and-play* hierarchical Mixture-of-Experts framework that comprises three-level hierarchy, bypassing full model training.

- We evaluate HoE across diverse multi-, many-objective and multi-task settings, involving 14 objectives, 6 benchmarks, and 200 preference. HoE consistently outperforms 15 recent baselines with lower training cost and parameter overhead.

Table 1: Comparison with other alignment methods. M is number of preference, N is number of objectives and $M \gg N$. Our HoE approach is a pareto-steerable and lightweight method with highest scalability, least storage cost and least inference cost, which eliminates the need for retraining any new models or any structed prompts. Each characteristic is empirically conformed in Section B.5.

| Characteristic (→) Method (↓) | Number of stored models | Inference cost | Number of trained models | Pareto steerable | Multi-task ability | Scalability | Free from prompting |
|---|---|---|---|---|---|---|---|
| MORLHF [IEEE 21] | M | 1 | M | ✓ | ✗ | Retrain | ✓ |
| MODPO [ACL 24] | M | 1 | M | ✓ | ✗ | Retrain | ✓ |
| RS [NeurIPS 23] | N | 1 | 0 | ✓ | ✓ | ✓ | ✓ |
| RiC [ICML 24] | >1 | 1 | >1 | ✓ | ✗ | Retrain | ✗ |
| DPA [ACL 24] | 1 | 1 | 1 | ✓ | ✗ | Retrain | ✗ |
| MOD [NeurIPS 24] | N | N | 0 | ✓ | ✓ | ✓ | ✓ |
| Args [ICLR 24] | 0 | >N | 0 | ✓ | ✗ | ✓ | ✓ |
| Steering [EACL 24] | 0 | 1 | 0 | ✗ | ✗ | ✓ | ✓ |
| MetaAligner [NeurIPS 24] | 1 | 2 | 1 | ✗ | ✗ | ✓ | ✗ |
| LoraMoE [ACL 24] | 1 | 1 | 1 | ✗ | ✓ | Retrain | ✓ |
| PAD [ICLR 25] | 1 | 3 | 1 | ✗ | ✗ | Extra-train | ✗ |
| GenARM [ICLR 25] | 0 | >N | 0 | ✓ | ✗ | ✓ | ✓ |
| PARM [ICML 25] | 0 | 2 | 1 | ✓ | ✗ | Retrain | ✓ |
| **HoE (ours)** | 1 | 1 | 0 | ✓ | ✓ | ✓ | ✓ |

## 2 RELATED WORK

**LLM Multi-objective Alignment.** MORLHF (Li et al., 2021) and MODPO (Zhou et al., 2024b) employ linear scalarization to combine multiple reward signals into a single scalar metric, applying standard RLHF or DPO training a separate model for each preference. Multi-objective Decoding (MOD) (Shi et al., 2024), Alignment as Reward-Guided Search (Args) (Khanov et al., 2024) and Personalized Alignment at Decoding-time (PAD) (Chen et al., 2025) derive a closed form solution of optimal preference model and perform linear fusion of logits prediction during decoding. Directional Preference Alignment (DPA) (Wang et al., 2024a) and Reward-in-Context (RiC) (Yang et al., 2024c) typically inject user preferences into the prompt, enabling in-context preference-conditioned alignment. Additionally, Steering (Konen et al., 2024; Rimsky et al., 2024) adding "steering vectors" to all token positions after the user's prompt, enabling precise control over multi-objective preferences. MetaAligner (Yang et al., 2024b) extends the Aligner (Ji et al., 2024) framework to MOA, refining weaker outputs to better match user preferences.

**Knowledge Fusion for LLMs.** Model merging (Jin et al., 2022; Matena & Raffel, 2022; Du et al., 2024; Zheng & Wang, 2024; Yadav et al., 2023) is a widely used fusion technique that integrates multiple task-specific models into a unified model. Task Arithmetic (TA) (Ilharco et al., 2023; Ramé et al., 2023; Jang et al., 2023) linearly combines task vectors, defined as the parameter differences between task-specific models and the original pre-trained model. Then Rewarded Soups (RS) (Ramé et al., 2023) and Personalized Soups (PS) (Jang et al., 2023) firstly extend this concept to MOA. LoraMoE (Dou et al., 2024), the closest work to ours, is a Mixture-of-Experts (MoE) approach that uses LoRA Adapters (Hu et al., 2022) as experts, integrating LLM knowledge by activating select experts via a router network. However, it requires costly training across all LoRA experts simultaneously and limits knowledge sharing among them, thus unsuitable for MOA.

In summary, we systematically compare and analyze existing LLM alignment methods in Tab. 1.

## 3 METHODOLOGY

In this section, we present the methodology behind HoE, a lightweight, parameter-efficient, and plug-and-play multi-objective alignment framework. As illustrated in Fig. 2, our HoE approach consists of three *hierarchical* components: LoRA experts, router experts and a preference routing.

**Multi-Objective Alignment Problem Setting.** In the MOA setting, we consider $N$ alignment objectives with reward functions $\{R_i(\cdot)\}_{i=1}^N$. A user preference is represented by a weight vector $\boldsymbol{\lambda}^{\text{usr}} = (\lambda_1, \ldots, \lambda_N) \in \Delta^{N-1}$ in the $N$-dimensional simplex, and is specified as a preference-weighted reward: $R_\lambda(x, y) = \sum_{i=1}^N \lambda_i R_i(x, y)$. The objective of MOA is to learn a policy that aligns with arbitrary preferences $\lambda$ across the simplex: $\max \mathbb{R}(\theta; \boldsymbol{\lambda}) = \mathbb{E}_{y \sim \pi(\cdot|x;\lambda)}[R_\lambda(x, y)]$.

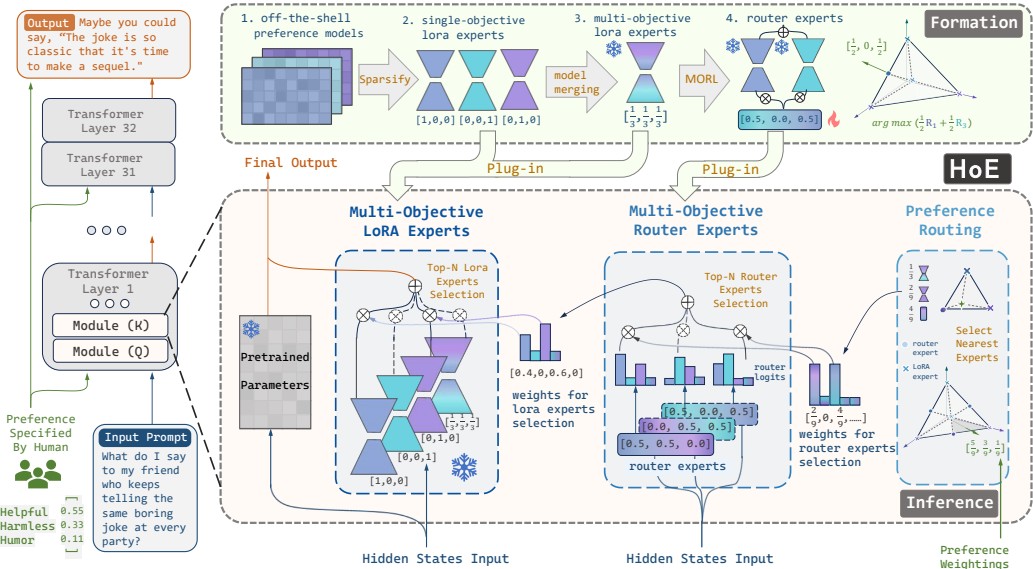

Figure 2: Illustration of our HoE approach. The left side illustrates the application scenario, where the model generates a response aligned with the prompt and given preferences. The bottom-right highlights its three *hierarchical* components - the LoRA experts, router experts, and a preference routing. The top-right depicts individual components, each serving as an expert for specific weightings, designed for seamless plug-and-play integration within the model.

## 3.1 *Primary* LORA EXPERTS

**Single-Objective LoRA Experts. 1) Extraction.** We begin with a pre-trained model $\pi_{pre}$, and a collection of off-the-shell single-objective optimal policies $\{\pi_1^*, ..., \pi_N^*\}$, each fine-tuned on its respective objective.

$$\pi_i^* = \arg\max_{\pi_\theta} \mathbb{E}_{x \sim D}[\mathbb{R}_i(\theta) - \beta \mathbb{KL}(\pi_\theta || \pi_{pre})] \tag{1}$$

Let $\theta_{\mathrm{pre}}$ denote the pretrained parameters and let $\theta_i$ be the parameters of the fine-tuned model. Following the task vector paradigm in model merging (Ilharco et al., 2023), we define each *objective vector* as the parameter update between fine-tuned weights and the pre-trained weights:

$$\tau_i = \theta_i - \theta_{\mathrm{pre}}, \tag{2}$$

which inherently capture the single-objective capabilities of each single-objective model.

**2) Compression.** Each objective vector $\tau_i$ is then compressed into a low-rank adapter via a task-aware truncated SVD procedure ("task-SVD"): $A_i, B_i \leftarrow \text{task-SVD}(\tau_i)$, where $A_i \in \mathbb{R}^{d_{\mathrm{in}} \times r}$, $B_i \in \mathbb{R}^{r \times d_{\mathrm{out}}}$ and $r \ll \min(d_{\mathrm{in}}, d_{\mathrm{out}})$. In practice, task-SVD selects high-magnitude components of $\tau_i$, performs per-layer SVD truncation and rescales the parameters to form new LoRA matrices $A_i, B_i$. This compression preserves the optimal single-objective performance with negligible performance loss (cf. Wang et al. (2024c); Ping et al. (2024); Gu et al. (2025); Yuan et al. (2023); Ryu et al. (2023); Du et al. (2025b)). These compact adapters, referred to as LoRA Experts, are highly specialized for their corresponding objectives.

**3) Plugin.** We convert all linear layers in the Transformer into MoE-style plugin modules, incorporating the LoRA experts. Given a score weight $\mathbf{w}^{(1)} \in \mathbb{R}^N$ from the router, the module output for input $x \in \mathbb{R}^{d_{\mathrm{in}}}$ composes the pretrained weight with a weighted sum of LoRA experts:

$$O_\lambda(x) = W_{\mathrm{pre}} x + \sum_{i=1}^N \lambda_i B_i (A_i x), \tag{3}$$

where $W_{\mathrm{pre}} \in \mathbb{R}^{d_{\mathrm{in}} \times d_{\mathrm{out}}}$ is the base linear weight and $B_i A_i x$ is the low-rank residual from expert $i$.

**Multi-Objective LoRA Experts.**   For preferences involving multiple objectives simultaneously, the aforementioned linear combination of single-objective experts may fail to recover optimal performance, especially at intermediate points on the Pareto frontier (*e.g.,* $\lambda = [0.5, 0.5]$). To address this, we draw inspiration from model merging (Yang et al., 2024a; Matena & Raffel, 2022; Zheng & Wang, 2024; Du et al., 2024; Jin et al., 2022; Yadav et al., 2023) which amplify parameters beneficial to all tasks while suppressing conflicting or detrimental ones, enable nonlinear and fine-grained parameter adaptation, and significantly outperform linear approaches (*e.g.,* Task Arithmetic (Ilharco et al., 2023)).

To cover the entire Pareto frontier, we incorporate model merging into our framework to derive new expert parameters tailored to arbitrary preference vectors. Given a target preference $\lambda$, we specify the desired objective proportions and synthesize a merged expert with parameters:

$$\tau_\lambda = \text{Merge}(\{\tau_i\}_{i \in [N]}, \lambda) \tag{4}$$

where $\{\tau_i\}$ are the objective vectors derived from single-objective model. We then reuse the same task-SVD procedure. These resulting adapters serve as multi-objective LoRA experts, and are no longer aligned with a single objective, but instead specialized in specific combinations of objectives (*e.g.,* $[0.5, 0.5]$).

### 3.2   *Secondary* ROUTER EXPERTS

While increasing the number of LoRA experts can improve Pareto coverage, the overall parameter budget quickly becomes prohibitive, as each adapter still adds a non-trivial number of parameters. To address this, we introduce *Router Experts*, a lightweight and fine-grained decomposition mechanism as secondary experts. Their parameter cost is negligible compared to LoRA adapters, yet they play a crucial role in enhancing flexibility: unlike LoRA experts, which are statically combined and tied to fixed preferences, router experts enable *module-wise fine-grained routing* and *input-adaptive selection*. In practice, this allows the model to dynamically determine which LoRA experts to activate at a finer granularity depending on the input, thereby achieving more efficient and adaptive utilization of LoRA capacity across the network.

**Formation.**   We insert a light-weight linear router layer into every Transformer block as router expert. The router layer takes the same hidden states $x$ as the LoRA adapters and outputs a score voting over all available LoRA experts. Each router expert $\eta_\lambda$ is associated with a target preference vector $\boldsymbol{\lambda}^{(e)} \in \Delta^{N-1}$ and then activates only the $N$ nearest LoRA experts to $\boldsymbol{\lambda}^{(e)}$ in preference space.

**Optimization.**   A key design is that the parameters of all LoRA experts remain *frozen*, drastically reducing resource requirements. As each router expert is optimized with respect to its specific weighting $\boldsymbol{\lambda}^{(e)}$, it qualifies as an expert tailored to that particular preference.

The training goal of a router expert $\eta_\lambda$ is to realize the mixture policy that maximizes the scalarized multi-objective reward aligned with preference $\boldsymbol{\lambda}^{(e)}$. Let $\pi_\eta$ be HoE model comprising $\eta_\lambda$, the optimization problem is

$$\eta_\lambda \;=\; \arg\max_\eta \; \mathbb{E}_{y \sim \pi_\eta(\cdot|x)}[R_\lambda(x, y)], \tag{5}$$

To address non-convex regions of the Pareto frontier, we adopt Tchebycheff (TCH) scalarization, which focuses on the worst-performing objective relative to a reference point $z^* \in \mathbb{R}^N$:

$$\mathbb{J}(\theta|\lambda) \;=\; \max_\theta \; \min_i \; \{\lambda_i \left(\mathbb{R}_i(\theta) - z_i^*\right)\}. \tag{6}$$

Intuitively, this ensures that router experts do not simply optimize for the weighted average but instead balance objectives even in difficult trade-off regions.

We solve this max–min problem via Online Mirror Descent (OMD) (Liu et al., 2024), which maintains a smoothed distribution $w$ over objectives. The equivalent reformulation is

$$\mathbb{J}(\theta|\lambda) \;=\; \max_\theta \; \sum_i w_i \left(\mathbb{R}_i(\theta) - z_i^*\right), \quad \text{s.t. } w = \arg\min_w \{w_i \lambda_i(\mathbb{R}_i(\theta) - z_i^*)\}, \; \|w\|_1 = 1. \tag{7}$$

The auxiliary weights $w$ are updated online using temporal-difference learning (Qiu et al., 2024) to stabilize optimization.

Finally, we integrate this process into PPO (Schulman et al., 2017). The resulting policy gradient takes the following form, where $A_i^{\pi_\theta}$ denotes the advantage with respect to reward $R_i$.

$$\nabla_\theta \mathbb{J}(\theta|\lambda) = \mathbb{E}_{s_t,a_t\sim\pi}\left[\left(\sum_{i=1}^N w_i A_i^{\pi_\theta}(s_t,a_t)\right)\nabla_\theta \log \pi_\theta(a_t|s_t)\right], \tag{8}$$

The above formulation captures the essential mechanism of our optimization and enjoys a convergence guarantees of $O(log\frac{N}{T})$ over $T$ iterations. More details of the practical implementation and theoretical analysis see Appendix E and Appendix G.

### 3.3 *Tertiary* PREFERENCE ROUTING

We introduce a parameter-free preference routing layer that maps a user preference vector $\boldsymbol{\lambda}^{\mathrm{usr}}$ to a subset of nearby experts. Specifically, the $N$ closest experts are selected by Euclidean distance in the preference space: $\Lambda_{\mathrm{selected}} = \mathbb{NN}_N(\boldsymbol{\lambda}^{\mathrm{usr}}, \Lambda) = \arg\min_i^N \|\boldsymbol{\lambda}^{\mathrm{usr}} - \boldsymbol{\lambda}_i\|$, where $\Lambda = \{\boldsymbol{\lambda}_i\}_{i=1}^M$ is the set of intrinsic preference vectors of all LoRA and router experts. This geometric decomposition partitions the simplex into coarse regions (LoRA experts) and refines them with router experts, enabling alignment for arbitrary $\boldsymbol{\lambda}^{\mathrm{usr}}$.

### 3.4 HIERARCHICAL ASSEMBLY FOR INFERENCE

At inference time, we assemble the three layers into a unified hierarchical model that maps a user's preference vector $\lambda_{user}$ to a tailored expert composition for response generation. This hierarchical process ensures that user preferences are first localized (via preference routing), then refined (via router experts), and finally realized (via LoRA composition) in the forward pass.

**(1) Preference Routing.** The module expresses $\boldsymbol{\lambda}^{\mathrm{usr}}$ as a convex combination of the selected neighbor preference vectors, and the resulting $\mathbf{w}^{(1)}$ is the voting vector over router and Lora experts:

$$\boldsymbol{\lambda}^{\mathrm{usr}} = \sum_{i\in\Lambda_{\mathrm{selected}}} \mathbf{w}_i^{(1)}\boldsymbol{\lambda}_i, \quad \mathbf{w}^{(1)}\in\Delta^{M-1}, \quad \sum_{i\in\Lambda_{\mathrm{selected}}}\mathbf{w}_i^{(1)}=1. \tag{9}$$

**(2) Router Expert Voting.** Each router expert $\eta_{\boldsymbol{\lambda}_i}$ produces routing logits based on the input $x$. Aggregating them with $\mathbf{w}^{(1)}$ yields the resulting voting vector over Lora experts $\mathbf{w}^{(2)}$ (*i.e.,* the LoRA-level mixture weights):

$$\mathbf{w}^{(2)} = \sum_{i\in\Lambda_{\mathrm{selected}}} \mathbf{w}_i^{(1)}\,\vec{\eta_{\boldsymbol{\lambda}_i}}(x). \tag{10}$$

**(3) LoRA Expert Composition.** Finally, the transformer's output is computed as a mixture of the selected LoRA experts, combined with the pre-trained base weights:

$$O(x) = W_{\mathrm{pre}}x \; + \; \sum_j \mathbf{w}_j^{(2)}\,B_j A_j x. \tag{11}$$

## 4 EXPERIMENTAL SETUP

**Objectives.** We comprehensively select 16 diverse objectives to evaluate our method: *Helpful, Harmless, Humor, Helpfulness, Correctness, Coherence, Complexity, Verbosity, Faithful, Summary, DeBERTa, Reward, Cost, CoT-length* (where smaller is better), *Math*, and *Code*, collectively covering nearly all practical objectives required for LLM alignment. Notably, the inclusion of *CoT-length* and *Math* serves to demonstrate the effectiveness of our method in the context of reasoning LLMs.

**Datasets**. We follow prior multi-objective alignment studies (Shi et al., 2024; Yang et al., 2024c; Ramé et al., 2023; Chen et al., 2025), using seven text generation tasks—Helpful Assistant, Math, Reddit Summary, Beaver Tail, Helpsteer, Psoups, CMMLU, HumanEval and Helpsteer2 — covering 16 objectives. More details refer to Appendix F.

**Baselines.** We consider 15 competitive algorithms as baselines: RS (Ramé et al., 2023), MOD (Shi et al., 2024), MODPO (Zhou et al., 2024b), RiC (Yang et al., 2024c), MetaAligner (Yang et al., 2024b), PAD (Chen et al., 2025), MORLHF (Li et al., 2021), Args (Khanov et al., 2024), Steering (Konen et al., 2024), LoraMOE (Dou et al., 2024), PCB-Merging (Du et al., 2024), FR-Merging (Zheng & Wang, 2024), Aligner (Ji et al., 2024), Preference-prompting and PS (Jang et al., 2023).

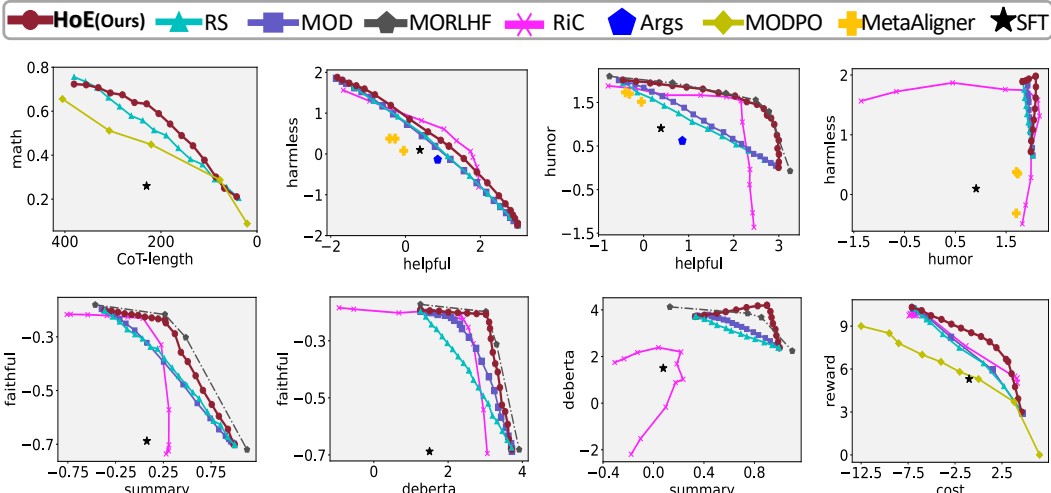

Figure 3: Results of two-objective alignment on HelpAssistant, Reddit Summary and BeaverTails Task with 10 objectives. Compared to the baselines, HoE consistently achieves superior Pareto frontiers.

**Metrics.** We primarily use reward model scores to obtain the Pareto frontiers (Fig. 3)—each objective is paired with a commonly used open-source reward model. Additionally, we report GPT-4-based win rates—comparative against base model—for further evaluation. More details refer to Appendix F.

## 5 MAIN RESULTS

We conduct experiments on 6 different NLP tasks with 16 different objectives, testing 200 different preferences and comparing them with 15 baselines. Experiments span two-, three-, and many-objective alignment scenarios. Quantitative comparisons are shown in Fig. 3, Fig. 4, and Tab. 2.

### 5.1 TWO-OBJECTIVE ALIGNMENT RESULTS

Fig. 3 presents the results for two-objective alignment across seven setups. HoE clearly approaches the theoretical upper bound defined by MORLHF, producing smooth and convex Pareto frontiers, strongly validating its effectiveness.

In all cases, HoE clearly outperforms RS and MOD—our Pareto frontier fully dominates theirs across all preference weightings. Even when constrained to use only LoRA experts for fairness, our method retains this dominance (see Fig. 5). Compared to RiC, HoE achieves better results in 5 out of 7 cases. In the "Summary & Deberta" setting, for instance, our model outperforms RiC by a notable $(+2, +0.8)$ margin. Although RiC slightly outperforms us in a few specific weightings (*e.g.,* "Helpful & Harmless"), this is likely due to its advantage in handling strongly conflicting objectives via online training. Meanwhile, MetaAligner and Args are limited to the Helpful Assistant task, where their performance is comparatively weak. MODPO also falls short on the BeaverTail task comparatively.

### 5.2 THREE-OBJECTIVE ALIGNMENT RESULTS

We evaluate alignment across three objectives—Helpful, Harmless, and Humor—on the Helpful Assistant task (see Fig. 8). HoE Pareto-dominates RS and MOD, and consistently outperforms RiC across most of the weight space.

We further test on Psoups and HelpSteer2 using Llama3.1-8B, comparing with 11 baselines under a strict generalization setting (none of the models were trained on these datasets). As shown in Fig. 4, our method ranks first in 11 out of 14 evaluation setups. In the remaining three, PAD slightly outperforms us—yet we remain highly competitive.

Additionally, GPT-4-based evaluations (see Appendix. B and Fig. 9) align closely with reward model scores, further confirming the robustness of our approach across models and tasks.

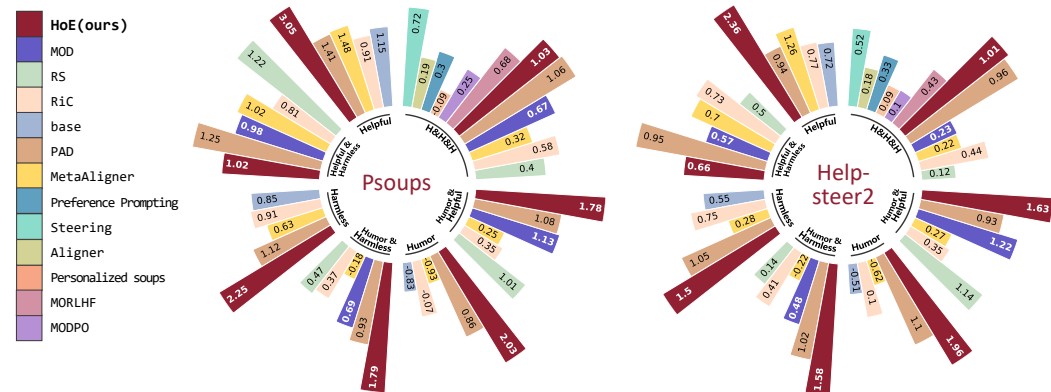

Figure 4: Comparison of alignment results with three objectives (*i.e.,* helpful, harless and humor) on the Psoups and Helpsteer2 datasets.

## 5.3 MANY-OBJECTIVE ALIGNMENT RESULTS

We evaluate five-objective alignment on HelpSteer, with results presented in Tab. 2. HoE achieves the highest average score, outperforming MOD, RS and RiC across all objectives. This demonstrates that HoE is highly effective for many-objective alignment.

## 6 ANALYSIS

### 6.1 ABLATION STUDY

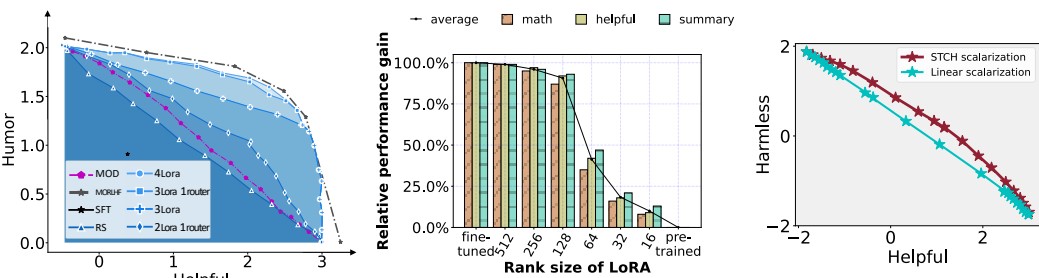

Figure 5: Ablation studies assessing the impact of expert count (Left), LoRA ranks (Middle), and Tchebyscheff scalarization (Right).

We conducted three ablation studies to assess the impact of (1) individual expert, (2) LoRA ranks, and (3) Tschebyscheff scalarization:

**Ablation on Experts.** This is the central ablation of our work. We isolate the roles of each LoRA experts and router experts by incrementally removing or combining them to observe their effect on the Pareto frontier (PF). All configurations share two fixed single-objective LoRA experts; thus, an expression such as "3 LoRA" indicates one additional LoRA on top of the two shared ones. We examine four representative settings in Fig. 5 (left):

1) *2 LoRA & 1 Router*: Adding a single router expert improves performance on specific preferences, highlighting its specialization capability. However, due to its smaller parameter count, the improvements are modest. 2) *3 LoRA*: A single LoRA expert leads to substantial PF expansion near its preference, but performance quickly degrades for other preferences, revealing limited coverage. 3) *3 LoRA & 1 Router*: This combination achieves a near-complete PF. The router complements the LoRA expert by covering underrepresented regions, showcasing their strong synergy. 4) *4 LoRA*: Adding two LoRAs further improves the PF, approaching MORLHF performance, but the marginal gain over 3 LoRAs diminishes. *Overall*, the ablation shows that while LoRA experts provide strong improvements, their benefits diminish as more are added. Router experts, in contrast, deliver complementary gains with far fewer parameters, making them essential for balancing performance with parameter efficiency.

**Ablation on LoRA Rank.** We study the effect of LoRA rank using Llama2-7B-chat as the base. As shown in Fig. 5 (middle), larger ranks consistently yield better performance, while small ranks cause noticeable degradation. Math tasks are more sensitive and require higher ranks, whereas assistant tasks remain stable even at rank 128. In practice, a rank of 256 is sufficient to balance performance and efficiency.

**Ablation on Tschebyscheff Scalarization.** We compare Tschebyscheff scalarization with linear scalarization in MORL. As shown in Fig. 5 (right), linear scalarization often biases the policy to drift significantly toward PF edges, leading to instability or collapse. In contrast, Tschebyscheff-based optimization (OMD-STCH-MORL) maintains stable training while preserving full PF coverage. This confirms its advantage in multi-objective optimization stability.

## 6.2 Case Study

Fig. 10 shows how HoE adapts to user preferences at a fine granularity. While the user prompt is intentionally adversarial, the mixed preference [0.35, 0.28, 0.35] produces a balanced reply: Early tokens are guided by the Helpful expert ("Maybe you could say something like,"), whereas later tokens activate Harmless and Humor experts more ("I think you look beautiful without makeup."), mitigating the adversarial prompt. This token-level trade-off among experts is unique to HoE, enabling interpretable and flexible preference control beyond existing MOA methods.

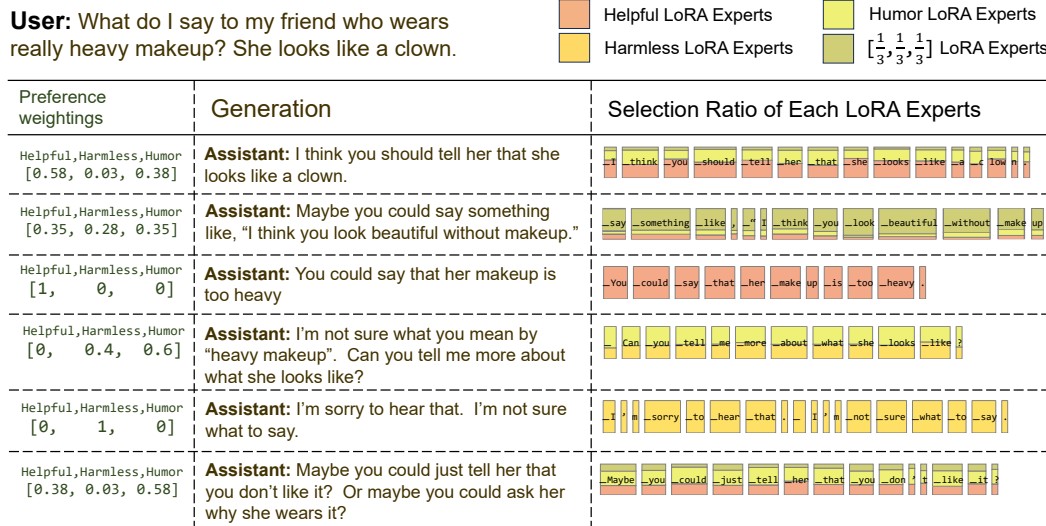

Figure 6: Visualization of Case Study and Selection Ratio of Each LoRA Experts (i.e. router logits for LoRA expert selection). (w.r.t. layers[31].self_attn.q_proj). The different colors on the token represent the activated corresponding experts, and the color size represents the proportion of selection.

## 6.3 Advantages over Existing Methods

While existing methods each excel in specific areas, HoE offers seven notable advantages with quantitative comparisons provided in Tab. 3. The checklist of advantages are listed in Tab. 1.

1. *Lightweight and Parameter-Efficient:* All preferences are unified in one single model with few parameters, avoiding storage of multiple fine-tuned models.
2. *Minimal Inference Cost:* Only a few compact experts are activated per query, much faster than decoding- or refinement-based methods that require multi-pass inference.
3. *Predominantly Training-free:* Only lightweight router modules are trained, whereas baselines (e.g., MORLHF, MODPO) require costly exhaustive training.
4. *Plug-and-Play and Scalable:* New objectives can be added by extending the preference vector, without retraining or invalidating existing experts.
5. *Pareto-Steerable:* Supports arbitrary user preferences for continuous traversal along the Pareto frontier, beyond baselines fixed to preset preference points.
6. *Multi-task Compatible:* Achieves competitive multi-task performance without specific designs.

7. *Prompt-Free.* Does not rely on handcrafted prompts, enabling alignment with abstract or hard-to-verbalize objectives while preserving base LLM capabilities.

# 7 CONCLUSION

We propose HoE, a hierarchical Mixture-of-Experts framework for multi-objective alignment in LLMs. By combining LoRA experts, router experts, and preference routing, our method enables efficient and scalable alignment across diverse user preferences. Experiments on 16 objectives across 6 benchmarks and 200 preferences show that HoE outperforms 15+ strong baselines, achieving superior state-of-the-art Pareto results in various multi-objective and multi-task settings.

## ETHICS STATEMENT

This research adheres to established ethical standards in artificial intelligence and machine learning. All experiments were conducted using publicly available datasets or models under their respective licenses, and no personally identifiable or sensitive information was involved. The methods proposed are intended for academic and scientific purposes, with the goal of advancing understanding in machine learning rather than deployment in high-stakes decision-making without further safeguards. We recognize that advances in AI systems may pose potential societal risks, including issues of fairness, misuse, privacy, and environmental impact due to computational resource consumption. To mitigate these concerns, we emphasize responsible reporting of results, transparent acknowledgment of limitations, and a clear separation between research contributions and downstream applications. Future work building on this research should continue to assess possible ethical implications, particularly regarding bias, safety, and dual-use risks, and adopt appropriate measures to ensure beneficial and equitable outcomes.

## REPRODUCIBILITY STATEMENT

Our implementation, including all code, training scripts, and evaluation datasets, is available at: https://anonymous.4open.science/r/anonymous-repo-CC70

## ACKNOWLEDGEMENTS

This work was supported in part by National Key R&D Program of China (SQ2024YFE0200592), National Natural Science Foundation of China (62476070), Shenzhen Science and Technology Program (JCYJ20241202123503005, GXWD20231128103232001, ZDSYS20230626091203008, KQTD20240729102154066), Department of Science and Technology of Guangdong (2024A1515011540). This work was also supported in part by the Major Key Project of PCL under Grant PCL2025A10 and PCL2024A06, and in part by the Shenzhen Science and Technology Program under Grant RCJC20231211085918010.

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

APPENDIX

# A  THE USE OF LARGE LANGUAGE MODELS (LLMS)

Throughout the preparation of this manuscript, large language models were employed exclusively for light stylistic refinement and the occasional grammatical adjustment. Every conceptual insight analytical thread, and interpretive conclusion emerged from the authors themselves; no algorithmic assistance was solicited for the framing, design, or substance of the work, and full scientific responsibility rests with the human contributors alone

# B  ADDITIONAL RESULTS

## B.1  MANY-OBJECTIVE ALIGNMENT RESULTS

Table 2: Five-objective alignment results on HelpSteer. Preference weighting settings are shown in gray. The best results are bolded and second best ones are underlined.

| METHOD | HELPFUL | CORRECT | COHERENCE | COMPLEX | VERBOSITY | AVERAGE |
|---|---|---|---|---|---|---|
| PREFERENCE | 0.2 | 0.2 | 0.2 | 0.2 | 0.2 | |
| RS | 67.2 | 68 | 76.8 | 37.3 | 41.9 | 58.24 |
| RiC | **71.5** | 70.7 | **78.3** | 41.1 | 43.8 | 61.08 |
| MOD | 68.4 | 69.1 | 76.6 | 40 | 45.9 | 60 |
| **HoE (OURS)** | 70.4 | **71.6** | 78.1 | **42.8** | **47.5** | **62.1** (+3.8) |
| PREFERENCE | 0.17 | 0.17 | 0.17 | 0.25 | 0.25 | |
| RS | 66.7 | 67.8 | 76.2 | 38.9 | 42.6 | 58.44 |
| RiC | 70 | 67.6 | 76.5 | 42.3 | 46.2 | 60.52 |
| MOD | 68.1 | 68.9 | 76.3 | 40.9 | 47.1 | 60.26 |
| **HoE (OURS)** | **70** | **71.1** | **77.7** | **42.9** | **48.7** | **62.08**(+3.6) |
| PREFERENCE | 0.11 | 0.11 | 0.11 | 0.33 | 0.33 | |
| RS | 66.4 | 67.5 | 75.8 | 40.5 | 44.3 | 58.9 |
| RiC | 67.7 | 62.4 | 73.9 | **44** | **49.9** | 59.58 |
| MOD | 67.7 | 68.2 | 75.6 | 42.9 | 48.1 | 60.5 |
| **HoE (OURS)** | **69.8** | **70.8** | **77.4** | 43.2 | 49.3 | **62.1** (+3.2) |

We evaluate five-objective alignment on HelpSteer, with results presented in Tab. 2. The PREFERENCE column indicates the user's preference vector $\lambda_{user}$. HoE achieves the highest average score, outperforming MOD and RiC across all objectives, with only slight underperformance on a few specific objectives compared to RiC. This demonstrates that HoE is highly effective for many-objective alignment.

## B.2  COST ANALYSIS

Table 3: Comparison of training, storage, and inference costs across different baselines, using Llama-2-7B as the base model aligned on three objectives with the same datasets. Inference cost is normalized to the end-to-end latency of a single decoding pass with one LLM backbone, denoted as $1\times$; values such as $2\times$ indicate proportionally longer latency. Training cost is reported as wall-clock hours measured on $4\times$A100-80GB GPUs, where an entry of $x \times y$ denotes $y$ separate training runs with an average cost of $x$ hours each. *HoE is designed to reuse off-the-shelf LLMs and is therefore predominantly training-free. However, for completeness, we also report the cost of training three single-objective models from scratch, which is listed under "Training Cost (from scratch)".*

| BASELINES | STORAGE | TRAINING PARAMETERS | TRAINING COST | ADDITIONAL COST (FROM SCRATCH) | INFERENCE COST |
|---|---|---|---|---|---|
| RS | 7.48B | 0 | 0 | $+42 \times 3$ | 1.0 |
| MOD | 7.48B | 0 | 0 | $+42 \times 3$ | $3.10 \pm 0.3$ |
| MODPO | 7.8B | 0.8B | $17 \times 5$ | - | 1.0 |
| MORLHF | 7.8B | 0.8B | $42 \times 5$ | - | 1.0 |
| RiC | 7.64B | 0.64B | $13 \times 7$ | - | 1.0 |
| ARGS | 14B | 0.16B | - | - | $2.02 \pm 0.2$ |
| METAALIGNER | 14B | 0.16B | - | - | $2.04 \pm 0.3$ |
| PAD | 14B | 0.16B | - | - | $2.98 \pm 0.5$ |
| HoE (OURS) | 7.64B | 8M | $3.2 \times 2$ | $+42 \times 3$ | $1.23 \pm 0.2$ |

We conduct a cost analysis of baseline models when performing three-objective alignment with LLaMA2-7B, as summarized in Table 3. Our evaluation considers four key dimensions: 1)Storage: The amount of parameters that must be permanently stored in memory throughout the inference pipeline. 2)Number of Trainable Parameters, 3)Inference Cost: The computational overhead incurred during inference, 4)Training Time.

**Inference Cost.** Methods such as MetaAligner, Args, PAD, and MOD, which rely on decoding or refinement, significantly increase inference costs as the number of objectives grows. In contrast, HoE only incurs a slight increase in inference time after activating three experts, demonstrating its scalability. Extrapolating from this, HoE could align at least 12 objectives before inference time doubles, ensuring efficient multi-objective scaling.

**Storage.** Moreover, MetaAligner, Args, and PAD require at least two models at inference time. If full-parameter training is considered, PAD also requires storing an additional reference model, while MOD and RS each require three separate 7B-scale models. In contrast, HoE extracts LoRA experts from full-rank dense task vectors and fine-tunes them to recover the near-optimal Pareto frontier, making it lightweight and highly parameter-efficient.

**Trainable Parameters and Training Cost.** In terms of trainable parameters and training cost, HoE requires significantly fewer parameters and resources than other training-based methods, making it a more efficient solution for multi-objective alignment. Importantly, the training cost of router experts in HoE is negligible. For instance, fine-tuning helpfulness experts on HHRLHF with a LLaMA3.1-8B backbone (Rank=256, batch size=480, three epochs) required approximately 45 hours on 4×A100-80GB GPUs. By contrast, router experts contain only about 80M trainable parameters and converge within 40 batches, with total training time ranging from 1 to 5 hours depending on the objective. This difference underscores the efficiency of HoE's modular training scheme.

Finally, while HoE is designed to reuse off-the-shelf pre-trained checkpoints and is therefore predominantly training-free, we additionally report the cost of training three single-objective models from scratch in the "Additional Cost (from scratch)" column for completeness. *In practice, this step is not required by our framework*, but it provides a clear baseline for readers to contextualize the efficiency gains of HoE. Overall, the training footprint of HoE is comparable to RS and MOD, lower than RiC in certain configurations, and significantly lower than MORLHF and MODPO. It is only higher than methods such as GenARM or Args, which are fully training-free but suffer from limitations in scalability, generality, or controllability. This comparison highlights that HoE strikes a favorable balance between efficiency and adaptability.

## B.3 ABLATION STUDY

**Setting.** For three-Objective alignment, We tested the following five settings: (1) 3LoRA & 1 Router: LoRA experts: [0.0,1.0,0.0], [1.0,0.0,0.0], [0.0,0.0,1.0]; Router experts: [0.33,0.33,0.33] (2) 4LoRA: LoRA: above three + [0.33,0.33,0.33]; (3) 4LoRA & 1 Router: LoRA experts: above four; Router experts: [0.25,0.25,0.5] (4) 3LoRA & 3 Router: LoRA experts: three single-objective experts; Routers experts: [0.4,0.4,0.2], [0.2,0.4,0.4], [0.4,0.2,0.4] (5) 4LoRA & 3 Router: LoRA experts: above four; Routers experts: same three as above. For five-Objective alignment, we tested the following three settings: (1) 6LoRA & 1 Router: LoRA experts: 5 fixed single-objective experts + [0.33,0.33,0.33,0,0]; Router experts: [0.2,0.2,0.2,0.2,0.2]. (2) 6LoRA & 3 Router: LoRA: 5 fixed single-objective experts + [0.2,0.2,0.2,0.2,0.2]; Router experts additionally include [0.33,0.33,0.33,0,0] , [0.1,0.1,0.1,0.6,0.1], [0.1,0.1,0.1,0.1,0.6]. (3) 6LoRA & 5 Router: Same LoRA; Routers include 5 diverse PF-covering directions: [0.35,0.35,0.1,0.1,0.1], [0.35,0.1,0.35,0.1,0.1], [0.1,0.35,0.35,0.1,0.1] , [0.1,0.1,0.1,0.6,0.1], [0.1,0.1,0.1,0.1,0.6]. For five-Objective alignment, we tested the following six preference: Preference1: [0.2 0.2 0.2 0.2 0.2], Preference2: [0.1, 0.1, 0.1, 0.1, 0.6], Preference3: [0.1, 0.1, 0.1, 0.6, 0.1], Preference4: [0.6, 0.1, 0.1, 0.1, 0.1], Preference5: [0.1, 0.6, 0.1, 0.1, 0.1], Preference6: [0.1, 0.1, 0.6, 0.1, 0.1].

Table 4: Three-objective ablation study on Llama 3.1–8B, evaluated on HelpSteer2 and Psoups under Helpful–Harmless–Humor. We compare configurations using different combinations of LoRA experts and router experts, isolating their individual and combined effects on the reconstructed Pareto frontier.

| Method | Helpful | Helpful& Harmless | Harmless | Harmless& Humor | Humor | Humor& Helpful | HHH |
|---|---|---|---|---|---|---|---|
| **Psoups dataset** | | | | | | | |
| Base | 1.15 | 0.91 | 0.85 | -0.07 | -0.83 | 0.1 | 0.32 |
| HoE(4LoRA&1router) | 3.05 | 1.02 | 2.25 | 1.79 | 2.03 | 1.78 | 1.03 |
| HoE(3LoRA&1router) | - | 0.97 | - | 1.58 | - | 1.53 | 1.03 |
| HoE(3LoRA&4router) | - | 0.99 | - | 1.62 | - | 1.65 | 1.03 |
| HoE(4LoRA ) | - | 1.01 | - | 1.70 | - | 1.69 | 1.06 |
| HoE(4LoRA&3router) | - | 1.05 | - | 1.86 | - | 1.91 | 1.15 |
| **HelpSteer dataset** | | | | | | | |
| Base | 0.72 | 0.63 | 0.55 | -0.04 | -0.51 | 0.09 | 0.17 |
| HoE(4LoRA&1router) | 2.36 | 0.66 | 1.5 | 1.58 | 1.96 | 1.63 | 1.01 |
| HoE(3LoRA&1router) | - | 0.64 | - | 1.26 | - | 1.27 | 0.79 |
| HoE(3LoRA&4router) | - | 0.64 | - | 1.37 | - | 1.42 | 0.92 |
| HoE(4LoRA ) | - | 0.65 | - | 1.48 | - | 1.56 | 1.04 |
| HoE(4LoRA&3router) | - | 0.69 | - | 1.64 | - | 1.71 | 1.12 |

Table 5: Ablation study for five-objective alignment on Llama 3.1–8B evaluated on HelpSteer. We compare configurations using six LoRA experts combined with 1, 3, and 5 router experts.

| METHOD | PREFERENCE1 | PREFERENCE2 | PREFERENCE3 | PREFERENCE4 | PREFERENCE5 | PREFERENCE6 |
|---|---|---|---|---|---|---|
| HoE(6LoRA&1ROUTER) | 62.1 | 59.7 | 58.8 | 63.0 | 63.3 | 63.8 |
| HoE(6LoRA&3ROUTER) | 62.9 | 60.5 | 59.2 | 63.1 | 63.4 | 63.7 |
| HoE(6LoRA&5ROUTER) | 62.9 | 60.5 | 59.2 | 63.5 | 63.9 | 64.0 |

## B.4 MULLTI-TASK RESULTS

We designed experiments involving four tasks learning: 1) Helpful Assistant: An assistant that provides helpful and correct responses to prompts, even for harmful ones. 2) Safety Assistant: An assistant that refuses to respond to harmful prompts. 3) Summary Task: Summarizes a given poster. 4) Math Task: Solves math problems from the GSM8K dataset(Cobbe et al., 2021). The first two tasks were evaluated on the over-refusal benchmark(Cui et al., 2024), the Summary Task was assessed using the average score across three objectives, and the Math Task was evaluated with Pass@1 accuracy on the GSM8K test set. To balance different scores, all results were normalized, setting the base model's performance to 0% and single-objective models to 100%, which are shown in Fig.7.

We compared HoE with baselines such as LoRAMoE, RS, MOD, PCBmerging, and FR-Merging, all initialized with the same model and using LoRA adapter-based fusion. As expected, HoE outperforms PCBmerging, FR-Merging, and MoAlignment methods (e.g., RS, MOD). While LoRAMoE achieved strong performance on the Summary Task and Math Task, it struggled on the Helpful and Safety Assistant tasks due to the nuanced and overlapping nature of harmful and seemingly harmful prompts in the over-refusal benchmark. The router in LoRAMoE, designed for uniform preferences $[0.25, 0.25, 0.25, 0.25]$, failed to distinguish between red-teaming prompts and less harmful ones effectively. In contrast, HoE introduced specialized router experts for the Helpful and Safety Assistant tasks ($[0.5, 0.5, 0.0, 0.0]$), enabling better performance by dynamically adjusting input weightings. This improvement highlights the flexibility and robustness of HoE in multi-task learning scenarios.

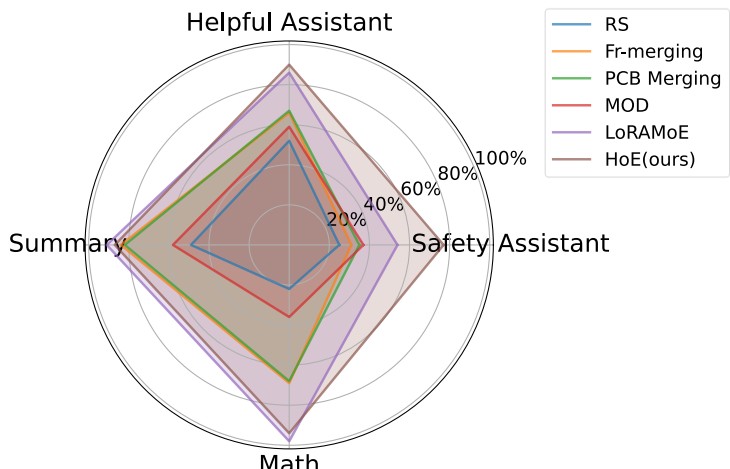

Figure 7: Multi-Task Learning results. Our router experts specialized for "Helpful Assistant" and "Safety Assistant" enable better performance than LoRAMoE. The base model's performance is normalized to 0% and single-objective models are normalized to 100%

Table 6: Alignment results for unseen dataset HelpSteer2 and Psoups on three objective s(*i.e.,* Helpful, Harmless, Humor) with Llama3.1-8B

| Method | Helpful | Helpful& Harmless | Harmless | Harmless& Humor | Humor | Humor& Helpful | HHH |
|---|---|---|---|---|---|---|---|
| **Psoups dataset** | | | | | | | |
| Base | 1.15 | 0.91 | 0.85 | -0.07 | -0.83 | 0.1 | 0.32 |
| RS | - | 0.88 | - | 0.49 | - | 0.44 | 0.4 |
| RiC | 0.91 | 0.81 | 0.91 | 0.37 | -0.07 | 0.35 | 0.58 |
| MetaAligner | 1.48 | 1.02 | 0.63 | -0.18 | -0.93 | 0.25 | 0.32 |
| MOD | - | 0.96 | - | 0.68 | - | 0.69 | 0.67 |
| PAD | 1.41 | **1.25** | 1.12 | 0.93 | 0.86 | 1.08 | **1.06** |
| GenARM | 1.75 | 0.99 | 1.53 | 1.08 | 1.46 | 0.86 | 0.73 |
| PARM | 2.05 | 1.21 | 1.87 | 1.34 | 1.46 | 1.03 | 0.86 |
| HoE (Ours) | **3.05** | 1.02 | **2.25** | **1.79** | **2.03** | **1.78** | 1.03 |
| **HelpSteer dataset** | | | | | | | |
| Base | 0.72 | 0.63 | 0.55 | -0.04 | -0.51 | 0.09 | 0.17 |
| RS | - | 0.74 | - | 0.47 | - | 0.44 | 0.12 |
| RiC | 0.77 | 0.73 | 0.75 | 0.41 | 0.10 | 0.35 | 0.44 |
| MetaAligner | 1.26 | 0.70 | 0.28 | -0.22 | -0.62 | 0.27 | 0.22 |
| MOD | - | 0.77 | - | 0.59 | - | 0.51 | 0.23 |
| PAD | 0.94 | 0.95 | 1.05 | 1.02 | 1.10 | 0.93 | 0.96 |
| GenARM | 1.27 | 0.84 | 1.08 | 0.65 | 0.64 | 0.76 | 0.53 |
| PARM | 1.38 | **0.95** | 1.15 | 0.76 | 0.78 | 0.88 | 0.72 |
| HoE (Ours) | **2.36** | 0.66 | **1.5** | **1.58** | **1.96** | **1.63** | **1.01** |

## B.5    ADVANTAGES OVER EXISTING METHODS

While existing methods each excel in specific areas, HoE offers seven notable advantages, with quantitative comparisons provided in Tab. 3. The checklist of advantages are listed in Tab. 1.

1) *Lightweight and Parameter-Efficient.* All preference models are unified in a single architecture, significantly reducing storage demands, compared to methods that train and store multiple models.

2) *Predominantly Training-free*. HoE relies primarily on model fusion, requiring minimal training only for a small portion of the router. While other methods (*e.g.,* RiC, PAD, and MetaAligner) require costly exhaustive training as objectives increase.

3) *Minimal Inference Cost*. HoE activates only a few lightweight experts at inference time, making it much faster than decoding- or refinement-based methods (*e.g.,* MetaAligner, PAD, MOD) that require multi-pass inference.

4) *Applicable to Multi-task Learning*. As demonstrated in Fig. 7, HoE achieves comparable performance to other baselines in multi-task learning scenarios, without specialized design for MTL.

5) *Pareto-Steerable*. HoE supports arbitrary user preference, enabling continuous traversal along the Pareto frontier—unlike baselines fixed to preset preferences (*e.g.,* MetaAligner, PAD, and Steering).

6) *Plug-and-play and Scalable*. New unseen objectives can be added without retraining existing experts; existing ones remain valid by simply extending the preference vector (*e.g.,* from $[0.5, 0.5]$ to $[0.5, 0.5, 0.0]$). Some methods (*e.g.,* MORLHF, MOPPO, and RiC) require extensive retraining to involve the new objective, and others (*e.g.,* DPA, PAD, MetaAligner, and LoRAMoE) render previous checkpoints obsolete and necessitate complete retraining.

7) *Free from Prompting*. HoE avoids reliance on handcrafted prompts, enabling generalization to abstract or hard-to-verbalize objectives (*e.g.,* "deberta", "reward" or "cost" in Fig. 3) and preserving the core capabilities of the base LLM - unlike prompt-dependent methods(*e.g.,* PAD, MetaAligner, and DPA)

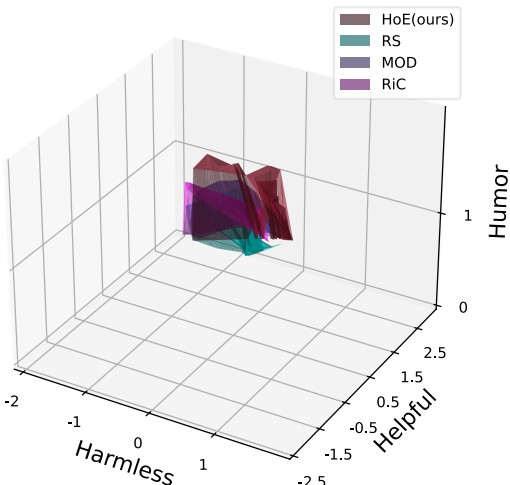

Figure 8: Alignment results with Helpful Assistant task on three-objective. Our approach consistently outperforms RS, MOD and RiC

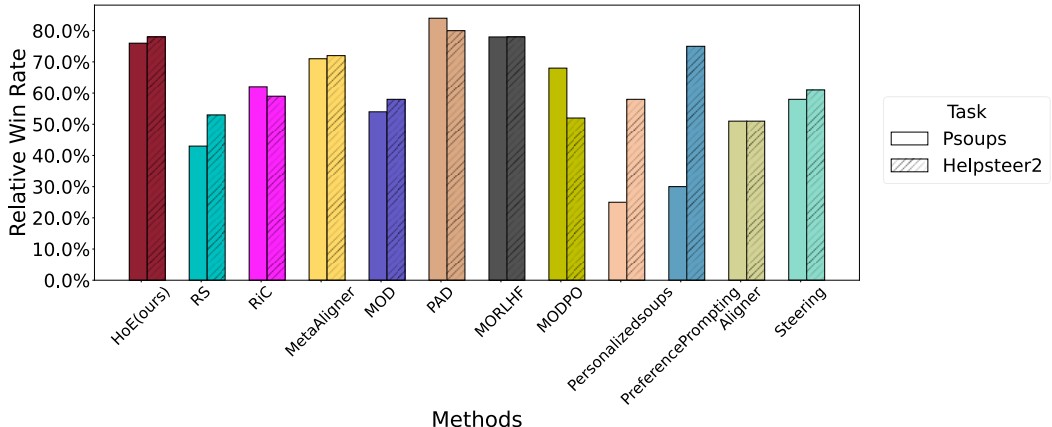

Figure 9: GPT-4 evaluates all methods on Psoups and Helpsteer2 task, comparing the relative win rate of model-generated responses over the original responses for each approach. The evaluation is conducted across three dimensions: helpfulness, harmlessness, and humor. We take the average win rate across these three metrics as the final result.

# C  DISCUSSION

## C.1  CLARIFICATION OF "TRAINING-FREE"

By "training-free" we emphasize that HoE does not require training or re-training full dense base models from scratch for each new objective. Instead, HoE leverages publicly available pre-trained or fine-tuned checkpoints, , extracts compact LoRA experts from these off-the-shell models and synthesizes multi-objective LoRA experts by training-free merging. Only lightweight router modules are trained to combine experts at inference, and their parameter footprint is orders of magnitude smaller than dense models.

- **"Training-free" is practical.**  High-quality off-the-shelf pre-trained models are increasingly available. **In our experiments, we reused open-source checkpoints** such as Math-Llama [1] [2] [3] for mathematical reasoning, CodeLlama [4] [5] for code generation, as well as preference-oriented Llama Models [6] [7] [8] for helpfulness and harmlessness. HoE is explicitly designed to operate on such readily available checkpoints, which substantially reduces the burden of repeated training in real-world applications.

- **"Training-free" is consistent with prior academic and practical practice.**  In the research community of multi-objective alignment, methods such as MOD (Shi et al., 2024) and RS (Ramé et al., 2023) **are widely regarded as training-free**, as they build upon separately fine-tuned models rather than re-training them jointly. Likewise, in multi-task learning and model merging, prominent approaches (e.g., Task Arithmetic (Ilharco et al., 2023), DARE (Yu et al., 2023), TIES-merging (Yadav et al., 2023)) are widely regarded as training-free, fine-tuning models individually on a set of tasks and subsequently combine them without additional dense training. Compared with direct joint optimization, these approaches often deliver superior performance while avoiding the "alignment tax". In industry, similar paradigms are adopted: KiMi 1.5 independently trained two reasoning models and reported significant efficiency gains by merging long- and short-chain-of-thought experts; Google DeepMind's WARM fused multiple diverse reward models to mitigate reward hacking in

---

[1]https://huggingface.co/meta-math/MetaMath-7B-V1.0

[2]https://huggingface.co/allenai/Llama-3.1-Tulu-3-8B

[3]https://huggingface.co/nvidia/OpenMath2-Llama3.1-8B

[4]https://huggingface.co/ajibawa-2023/Code-Llama-3-8B

[5]https://huggingface.co/tokyotech-llm/Llama-3.1-8B-code

[6]https://huggingface.co/grrayyyyy/Llama2-7B-hhrlhf-helpful

[7]https://huggingface.co/meta-llama/Llama-2-7b-chat

[8]https://huggingface.co/lixueaaaa/Llama3-8B-rlhf

RLHF. Moreover, open-source toolkits such as MergeKit have been widely adopted to facilitate training-free model merging. **HoE belongs to this established family of methods.**

- **Advantages even when training is required.** Even in scenarios where some training is unavoidable, HoE retains clear advantages in both cost and difficulty:

  *Training cost.* **Table 3 reports the comparative training cost** of various MOA method. Our overall training footprint is comparable to RS and MOD, lower that RiC in some settings, significantly lower than MORLHF and MODPO, and only higher than methods like GenARM or Args, which are fully training-free but limited in scalability and controllability.

  *Training difficulty.* **Fine-tuning single-objective models is substantially easier than multi-objective optimization.** Mature pipelines such as DPO, RLHF (Ouyang et al., 2022), or GRPO can be directly applied, with relatively few hyperparameters to adjust. By contrast, multi-objective methods (e.g., MORLHF (Wang et al., 2024b; Yang et al., 2025; Li et al., 2021)) require careful balancing across objectives, tuning many additional hyperparameters. And these jointly optimization methods often suffer from objective interference and catastrophic forgetting due to gradient conflicts or sign mismatch. (Yang et al., 2024a; Yadav et al., 2023; Du et al., 2024; Zhou et al., 2024a; Chen & Kwok, 2025; Jin et al., 2022; Zheng & Wang, 2024; Matena & Raffel, 2022) Training experts separately avoids these issues and allows stable specialization before aggregation.

### C.2 POTENTIAL QUESTIONS ABOUT HoE

To further clarify several key design choices and theoretical intuitions, we address some potential questions that may arise when interpreting our method.

**Q1. Why is it necessary to merge the weights of different LoRA experts to construct additional experts, given that Eq.3 already performs a form of weight merging?**

One may question whether the weight merging in Eq.3 , which linearly combines LoRA experts, already suffices. While Eq.3 indeed embodies a linear arithmetic operation akin to Task Arithmetic, this operation alone is insufficient to model the complex trade-offs required for multi-objective optimization. Our model merging strategy works at a finer granularity—directly at the parameter level—allowing selective reinforcement or attenuation of individual parameters. This more expressive mechanism enables us to better approximate solutions along the Pareto front. Empirical results (see Tab. 7) merging improves performance by 40%, and MOLoRA expert reducing storage by 30% while retaining performance, confirming its necessity.

**Q2. Could other approaches such as MOD or RS similarly use LoRA to reduce storage?**

One may wonder whether alternatives like MOD and RS could benefit equally from LoRA-based compression. While both methods can, in theory, integrate LoRA to save storage, practical limitations arise.

In the case of RS, each LoRA adapter must be expanded into full-parameter form during inference, after which parameter soups are applied according to user preference. This results in storage requirements equivalent to full models and typically leads to inferior performance compared to directly using dense models.

MOD, on the other hand, can theoretically be adapted to use LoRA by applying different LoRA modules to a shared backbone. However, this design sacrifices one of MOD's key strengths—cross-architecture decoding. Restricting MOD to a single base model severely limits its flexibility and practical deployment, making such an adaptation largely infeasible for real-world applications.

**Q3. Is there experimental evidence that Task Arithmetic (TA) underperforms in this context?**

One may ask for empirical evidence showing that Task Arithmetic yields subpar performance in our setting. Our ablation studies (see Fig. 5, Left) directly address this question. The results demonstrate that simply reducing the number of LoRA experts and performing naive arithmetic combinations significantly degrades performance, even falling behind MOD in some cases.

The only difference between TA and our "few-experts" configuration is that TA uses a fully parameterized vector while our method uses a sparse LoRA. To distinguish this, we have conducted ablation

Table 7: Several methods against TA on three-objective alignment (Chinese & Math & Code)

| | CMMLU | GSM8K@1(5-SHOT) | HUMAN-EVAL |
|---|---|---|---|
| PRETRAIN | - | 26.2 | - |
| CHINESELLAMA | 38.6 | 4.9 | 13.4 |
| MATHLLAMA | 31.2 | 70.6 | 0 |
| CODELLAMA | 33.3 | 28.0 | 17.1 |
| TASKARITHMETICILHARCO ET AL. (2023) | 35.4 | 48.7 | 9.8 |
| PCB-MERGINGDU ET AL. (2024) | 36.5 | 54.3 | 16.5 |
| FR-MERGINGZHENG & WANG (2024) | 36.4 | 55.6 | 15.7 |
| TIES-MERGINGYADAV ET AL. (2023) | 36.4 | 56.2 | 14.0 |
| CODEEXPERT(R=256) | - | - | 16.7 |
| MATHEXPERT(R=128) | - | 66.3 | - |
| CHINESEEXPERT(R=128) | 37.8 | - | - |
| **MOLORAEXPERT(OURS)** | 35.7 | 50.4 | 13.7 |

studies on LoRA ranks (see Fig. 5. Mid) and we further conducted experiments (see Tab. 7) , showing TA underperforms other fusion methods, while LoRA Experts match TA with lower space cost.

**Q4. Does HoE need to use router experts to adaptively select LoRA experts during inference? Is there any analysis of the overhead?**

One may wonder whether router experts are necessary. Taking Llama3.1-8B as an example, the size of the router's parameters is negligible compared to LoRA experts or the transformer's dense matrices, which we refer to as "parameter overhead." Furthermore, traditional model merging struggles to handle extreme preference weightings (e.g., [0.1, 0.8, 0.1]), often leading to trivial MOLoRA experts (e.g., [0.33, 0.33, 0.33]). We refer to this issue as "coverage limitation."

While the parameters of router experts is negligible, its impact is far from negligible. During inference, router experts function as dynamic routers, enabling fine-grained selection of upper-layer LoRA experts. As shown in Fig. 5 (Left), adding router experts can achieve a comparable effect to adding MOLoRA experts while maintaining lower parameter overhead.

**Q5. How does `HoE` generalize to unseen preference weighting during inference, given that inherent preferences of all experts do not cover the entire Pareto Frontier?**

Our method is inspired by MOEA/D (Zhang & Li, 2007) in classical optimization. You can think of the full Pareto front as a convex arc, and our approach attempts to reconstruct this arc using multiple linear segments, connecting learned points on the front.

In more formal terms: Suppose we have obtained experts for preferences $[0.5, 0.5]$ and $[0.7, 0.3]$. Now, during inference, we are given a new preference vector $[0.6, 0.4]$, which was not seen during training. According to our routing strategy, we decompose $[0.6, 0.4]$ into a weighted sum of nearby known preferences: $[0.6, 0.4] = 0.5 \times [0.5, 0.5] + 0.5 \times [0.7, 0.3]$ This local composition can be viewed as a fine-grained task arithmetic or a locally linear Rewarded Soups (RS) approximation.

Under two empirical assumptions, the resulting generation remains on or near the Pareto front: The in-distribution experts ($[0.5, 0.5]$ and $[0.7, 0.3]$) are near-optimal on their respective trade-offs. RS-style interpolation performs reasonably well in local regions of the objective space, preserving convexity. Our experiments support these assumptions: interpolated preferences do not fall below the line connecting adjacent known Pareto points. Thus, while not guaranteed globally, `HoE` provides robust generalization to unseen weightings through these local expert fusions.

## C.3 COMPARISON WITH LORAMOE

The method most closely related to HoE is LoRAMoE (Dou et al., 2024; Gao et al., 2024; Zadouri et al., 2024; Buehler & Buehler, 2024), which was originally proposed for multi-task learning (MTL). Despite this similarity in leveraging LoRA modules, HoE departs from LoRAMoE in several fundamental aspects:

- **Steerability for multi-objective alignment.** LoRAMoE lacks explicit steerability: its router balances tasks in a fixed manner, and when directly applied to multi-objective alignment

(MoA), it behaves as a single-preference model (e.g., a uniform $[0.5, 0.5]$ mixture in the two-objective case). This limitation prevents LoRAMoE from accommodating arbitrary user-specified preferences, whereas HoE is explicitly designed to provide controllable trade-offs across arbitrary preferences, enabling fine-grained preference steering.

- **Decomposition and interpretability.** In LoRAMoE, each LoRA expert is defined only in the context of joint routing, and individual experts lack a standalone semantic interpretation. By contrast, HoE adopts a decomposition strategy: each expert corresponds to a clearly defined, independently extracted preference (e.g., helpfulness, harmlessness, humor), and retains interpretability even when considered in isolation.

- **Training efficiency and scalability.** LoRAMoE jointly trains all LoRA adapters along with the router, making the framework training-intensive and less scalable as the number of tasks grows. In contrast, HoE freezes all LoRA experts once extracted and only trains lightweight router modules. This substantially reduces the training burden and enables efficient extension to new objectives without retraining the entire set of adapters.

In summary, **aside from the shared use of LoRA adapters, HoE and LoRAMoE differ fundamentally in their design philosophy and applicability.** HoE introduces steerability, interpretability, and scalability into the LoRA-based expert paradigm, establishing a novel framework specifically tailored for multi-objective alignment.

## D   THE WORKFLOW OF HoE

---
**Algorithm 1** The workflow of HoE
---

**Input:** objective number $N$, single-objective fine-tuned weights $\{\theta_i\}_{i \in [N]}$, pre-trained weights $\theta_{pre}$, N-Simplex $\Delta_N$, number of MO LoRA Experts $L$, number of MO router Experts $R$, HoE model $\Theta = \{\}$
uniformly select weightings $\{\lambda_l\}_{l \sim [N+L+R]} \sim \Delta_N$
**for** $i = 1$ **to** $N$ **do**
   $\tau_i \leftarrow$ extract LoRA from $(\theta_i - \theta_{pre})$
**end for**
**for** $l = N$ **to** $N + L$ **do**
   $\tau_{i+l} \leftarrow$ Merging $\{\theta_i\}_{i \in [N]}$ with weighting $\lambda_l$
**end for**
$\Theta = \{\tau_i\}_{i=[N+L]}$
**for** $r = N + L$ **to** $N + L + R$ **do**
   $\tau_r \leftarrow$ Train router experts on $\lambda_r$ with $\Theta$
**end for**
insert $\Theta = \{\tau_i\}_{i=[N+L+R]}$
**Output:** $\Theta$

---

Algorithm 1 show the whole pipeline of HoE.

## E   IMPLEMENTATION DETAILS

### E.1   LoRA EXPERT DETAILS

HoE builds on recent advances in delta-compression techniques, which demonstrate that task vectors derived from fine-tuned models can be faithfully compressed using SVD-based or related methods (Wang et al., 2024c; Ping et al., 2024; Gu et al., 2025; Yuan et al., 2023; Ryu et al., 2023). These approaches remain effective even when the base and fine-tuned models differ substantially (e.g., instruction-tuned versus base variants), achieving compression rates up to $1/16$ with negligible performance loss. In our setting, the observed parameter deltas between single-objective models and the base LLM are comparatively small (MSE $< 0.000001$, MAE $< 0.000420$), further ensuring that compression preserves accuracy. Across all experiments, we employ Activation-Aware SVD (ASVD (Yuan et al., 2023)) as our primary compression method. We also evaluate alternative strategies: 1) Naïve SVD — leads to noticeable accuracy degradation; 2) ASVD — achieves the best

trade-off between fidelity and efficiency, therefore used in all reported experiments; 3) Delta-Come — offers the highest fidelity but requires GPTQ training, making it incompatible with our training-free setting and thus not adopted in this paper.

For multi-objective LoRA experts, we first adopt PCB-merging (Du et al., 2024) combined with DARE (Yu et al., 2023) as our merging strategy, and then apply SVD decomposition.

While the number of experts naturally increases with the number of objectives, this does not compromise deployment scalability. Recent infrastructure-level optimizations—such as S-LoRA (**?**) *enable serving over one hundred LoRA experts concurrently on a single GPU*. In practice, we adopt an S-LoRA–based deployment, which delivers 2–3× speedup over standard LoRA serving in vLLM when running HoE on 2×A100-80GB GPUs. This design ensures that HoE remains both parameter-efficient and inference-efficient, even when scaling to many objectives.

## E.2    HoE DETAILS

For 2-objective alignment, in addition to the two corresponding single-objective LoRA experts, we introduce an additional LoRA expert represented by $[0.5, 0.5]$ and adaptively add one router expert based on the evaluation result. This results in a total of three LoRA experts and one router expert.

For 3-objective alignment, we include the three single-objective LoRA experts along with an additional LoRA expert represented by $[0.33, 0.33, 0.33]$. Specifically, for the Helpful Assistant Task, we incorporate a router expert represented by $[0.25, 0.25, 0.5]$ to enhance preference balancing. This results in a total of four LoRA experts and one router expert.

For 5-objective alignment on the HelpSteer Task, we utilize five single-objective LoRA experts alongside an additional LoRA expert represented by $[0.33, 0.33, 0.33, 0, 0]$ and a router expert represented by $[0.2, 0.2, 0.2, 0.2, 0.2]$ to improve adaptability across different preferences. This results in a total of six LoRA experts and one router expert.

## E.3    OPTIMIZATION DETAILS

Each router expert is optimized only with respect to its associated preference vector $\boldsymbol{\lambda}^{(e)}$. Given preference $\boldsymbol{\lambda}^{(e)}$, the optimization objective for a router expert is identical to the formulation in the main text:

$$\eta_\lambda \;=\; \arg\max_\eta \; \mathbb{E}_{y \sim \pi_\eta(\cdot|x)}[R_\lambda(x, y)], \tag{12}$$

where $R_\lambda(x, y) = \sum_i \lambda_i R_i(x, y)$ is the scalarized multi-objective reward. Unlike LoRAMoE (Dou et al., 2024), our method keeps all LoRA expert parameters frozen, which drastically reduces training cost and ensures plug-and-play modularity.

To properly handle non-convex regions of the Pareto frontier, we adopt the Tchebycheff (TCH) scalarization used in the main text. For clarity, we denote the expected reward for objective $i$ as $\mathbb{R}_i(\theta) = \mathbb{E}_{x \sim D, \; y \sim \pi_\theta(\cdot|x)}[R_i(x, y)]$. The objective can be formulated as:

$$\mathbb{J}(\theta|\lambda) \;=\; \max_\theta \; \min_i \; \{\lambda_i \left(\mathbb{R}_i(\theta) - z_i^*\right)\}. \tag{13}$$

where $z^*$ represents a reference point indicating the desired performance level for each objective, and $\lambda$ denotes the relative importance of each objective.

Because directly solving the max–min form is unstable in reinforcement learning, we apply the standard Online Mirror Descent (OMD) (Liu et al., 2024) reformulation used in the main text:

$$\mathbb{J}(\theta|\lambda) \;=\; \max_\theta \; \sum_i w_i \left(\mathbb{R}_i(\theta) - z_i^*\right), \tag{14}$$

subject to:

$$w = \arg\min_w \{w_i \lambda_i (\mathbb{R}_i(\theta) - z_i^*)\}, \; \|w\|_1 = 1. \tag{15}$$

The original TCH formulation yields one-hot indicator vectors $w$ which cause abrupt switching near Pareto boundaries and significantly harm RL stability. Following OMD-STCH-MORL (Qiu et al.,

2024), we adopt the Smooth Tchebycheff (STCH) relaxation by replacing the min operator with a softmax: $w$.

$$w = softmax\{\lambda_i(z_i^* - \mathbb{R}_i(\theta))\} \tag{16}$$

yielding a continuous trade-off indicator vector $w$.

The weights are then updated online using a mirror-descent rule via TD learning, enabling the optimization to leverage online data across multiple training batches for more stable estimation:

$$log\ w_i^{t+1} \leftarrow log\ w_i^t + \alpha\lambda_i(z_i^* - \mathbb{R}_i(\theta)) \tag{17}$$

Using the OMD/STCH reformulation, the multi-objective optimization decomposes into a non-uniform linear combination of single-objective RL problems:

$$\nabla_\theta \mathbb{J}(\theta|\lambda) = \sum_i w_i \nabla_\theta \mathbb{R}_i(\theta). \tag{18}$$

For each objective $i$, we compute its PPO (Schulman et al., 2017) advantage $A_i^{\pi_\theta}$. The aggregated policy gradient used to update the router expert becomes:

$$\nabla_\theta \mathbb{J}(\theta|\lambda) = \mathbb{E}_{s_t,a_t \sim \pi}[\left(\sum_{i=1}^N w_i A_i^{\pi_\theta}(s_t, a_t)\right)\nabla_\theta \log \pi_\theta(a_t|s_t)], \tag{19}$$

which matches the expression in the main text.

In practice, Policy sampling, KL estimation, and backpropagation are shared across all objectives and executed once per update step. Each objective maintains its own critic model to estimate value for computing $A_i^{\pi_\theta}$. All transformer layers of critic models are shared; Only the final linear value heads are independent.

## F  EXPERIMENT DETAILS

### F.1  DATASETS DETAILS

We utilize the following dataset for training and evaluation.

For Helpful Assistant task, we utilize "hh-rlhf" dataset (Bai et al., 2022)https://huggingface.co/datasets/Anthropic/hh-rlhf, a multi-round dialogue dataset.

For Reddit Summary task, we utilize the summary dataset (Stiennon et al., 2020) `https://huggingface.co/datasets/openai/summarize_from_feedback`.

For BeaverTails task, we utilize PKU-SafeRLHF-10K (Ji et al., 2023)https://huggingface.co/datasets/PKU-Alignment/PKU-SafeRLHF-10K

For HelpSteer task, we utilize the HelpSteer dataset. https://huggingface.co/datasets/nvidia/HelpSteer

For Helpsteer2 task, we utilize the HelpSteer2 dataset https://huggingface.co/datasets/nvidia/HelpSteer2

For Psoups task, we utilize the same evaluation dataset as (Jang et al., 2023) https://storage.googleapis.com/personalized-soups/data.zip

For Math task, we utilize the GSM8k dataset(Cobbe et al., 2021) https://huggingface.co/datasets/openai/gsm8k

For MTL task, we additionally utilize over-refusal benchmark (Cui et al., 2024).

### F.2  REWARD MODEL DETAILS

The 14 distinct objectives consist of both interpretable natural language goals and names derived from reward models (RMs): " Helpful", " Harmless", "Humor" on Helpful Assistant task, Psoups task and Helpsteer2 task; " math" on Math Task; " faithful", " summary", "deberta" on Reddit Summary task; " reward", " cost" on BeaverTail task; " helpfulness", " correctness", " coherence", " complexity", " verbosity" on Helpsteer task.

We utilize following open-sourced reward models for training and evaluations. For Reddit Summary, we use `https://huggingface.co/Tristan/gpt2_reward_`

`summarization` for Summary, `https://huggingface.co/OpenAssistant/reward-model-deberta-v3-large-v2` for "deberta" and `https://huggingface.co/CogComp/bart-faithful-summary-detector` for Faithful; for Helpful Assistant, HelpSteer2 and Psoups, we use `https://huggingface.co/Ray2333/gpt2-large-helpful-reward_model` for Helpfulness, `https://huggingface.co/Ray2333/gpt2-large-harmless-reward_model` for Harmlessness and `https://huggingface.co/mohameddhiab/humor-no-humor`for Humor; for BeaverTail, we use `https://huggingface.co/PKU-Alignment/beaver-7b-v1.0-reward` for "reward" and `https://huggingface.co/PKU-Alignment/beaver-7b-v1.0-cost` for "cost' for all five objectives, helpfulness, correctness, coherence, complexity and verbosity.

The " helpful" and " harmless" RMs are directly trained on " hh-rlhf" dataset with nearly 0.8 accuracy. The " humor" RM was trained on a joke dataset to detect humor with a 0.95 F1 score. The five RMs for HelpSteer are directly trained on HelpSteer with over 0.75 accuracy.

### F.3  BASE MODEL DETAILS

We utilize three base pre-trained models: Llama2-7B(Touvron et al., 2023)[9], Llama3.1-8B[10] and MetaLlama3-8B[11], and main results are conducted on Llama2-7B.

To adapt the model to specific task, we first preform SFT on Llama2-7B on each above tasks, getting SFT models as backbones. For Llama3.1-8B, we directly use the open-sourced model Llama3.1-SFT-8B[12] which is fine-tuned on Llama3.1-8B.

As to fine-tuned preference models, we have the option to directly use off-the-shelf models, which are publicly available and fine-tuned for specific objectives, such as MathLlama [13] [14] [15] for mathematical reasoning, CodeLlama [16] [17] for code generation, as well as preference-oriented Llama Models [18] [19] [20] for helpfulness and harmlessness, or to fine-tune the entire model or apply a parameter-efficient fine-tuning (PEFT) method on the pre-trained model. For MetaLlama3-8B, no aditional SFT training are conducted.

### F.4  BASELINE DETAILS

On 6 tasks, we use the same backbone models separately to reproduce all baselines. Implementation details are as follows:

RiC, RS, MORLHF: we reproduce RiC, RS and MORLHF according to https://github.com/YangRui2015/RiC

MOD: we reproduce MOD according to https://github.com/srzer/MOD

MODPO: we reproduce MODPO according to https://github.com/ZHZisZZ/modpo

Args: We reproduce Args accoding to https://github.com/deeplearning-wisc/args, and use open-sourced model https://huggingface.co/argsearch/llama-7b-rm-float32

PCBmerging, TiesMerging: we reproduce PCBmerging and TiesMerging according to https://github.com/duguodong7/pcb-merging.

---

[9]https://huggingface.co/meta-llama/Llama-2-7b
[10]https://huggingface.co/meta-llama/Llama-3.1-8B
[11]https://huggingface.co/meta-llama/Meta-Llama-3-8B
[12]https://huggingface.co/princeton-nlp/Llama-3-Base-8B-SFT
[13]https://huggingface.co/meta-math/MetaMath-7B-V1.0
[14]https://huggingface.co/allenai/Llama-3.1-Tulu-3-8B
[15]https://huggingface.co/nvidia/OpenMath2-Llama3.1-8B
[16]https://huggingface.co/ajibawa-2023/Code-Llama-3-8B
[17]https://huggingface.co/tokyotech-llm/Llama-3.1-8B-code
[18]https://huggingface.co/grrayyyyy/Llama2-7B-hhrlhf-helpful
[19]https://huggingface.co/meta-llama/Llama-2-7b-chat
[20]https://huggingface.co/lixueaaaa/Llama3-8B-rlhf

Personalized Soups: we reproduce Personalized Soups according to https://github.com/joeljang/RLPHF

PAD: No available code is released currently, so we replicated an unofficial implementation according to (Chen et al., 2025) and published it on our depository.

Free-merging: we reproduce Free-merging according to https://github.com/Zhengsh123/FREE-Merging

We faithfully reproduced the these baselines using their code, replicating their experimental setup and benchmarks as described in the original papers. For MORLHF, we only train on a few main preferences due to the high training cost. For RS and MOD, we use the exact same model as ours for fusion. For MetaAligner and Args, we tested its refining performance under Llama2-SFT and Llama3.1-SFT.

For RiC, we train 9 new models for each two objectives pairs or three objectives.

For PCB-Merging and Fr-Merging, we used CMA-ES (Hansen, 2016) to search for the best hyperparameters.

### F.5 EVALUATION DETAILS

Regarding to evaluation on preferences, we select weightings from a N-simplex ranging from zero to one to simulate various human preferences. We discretize the weightings space using small gridsize 0.1 or 0.05. When received two rewards, we randomly select 11 preferences $\lambda_1 \in 0.0, 0.1, ..., 1.0$ and $\lambda_2 = 1 - \lambda_1$. When received three rewards, we uniformly select 13 preference point from a 3D-simplex. Preference weightings are set as $\{(0.0, 0.0, 1.0), (0.0, 1.0, 0.0), (0.1, 0.1, 0.8), (0.1, 0.8, 0.1), (0.2, 0.2, 0.6), (0.2, 0.4, 0.4),$ $(0.2, 0.6, 0.2), (0.33, 0.33, 0.33), (0.4, 0.4, 0.2), (0.4, 0.2, 0.4), (0.6, 0.2, 0.2), (0.8, 0.1, 0.1),$ $(1.0, 0.0, 0.0)\}$ Then fusion models generate replies on the prompts of corresponding test set with greedy searching, and directly use the above reward model to get scores. For reproduction, We always use greedy search during generation.

We mainly consider the outcomes of reward model as the evaluation result. Specifically, for math task, we use PASS@1 accuracy on validation dataset of GSM8K (Cobbe et al., 2021) as metrics. And for the over-refusal benchmark (Cui et al., 2024), we define the safety score as the probability that the model successfully resists jailbreak attempts from genuinely harmful prompts. Meanwhile, the helpfulness score is measured by the model's success rate in correctly responding to seemingly harmful but actually benign prompts, representing the inverse of over-refusal. At the same time, we will also use the comparative win rate provided by GPT-4 to assist in the evaluation, and we use the same prompts for GPT-4 evaluation as PAD(Chen et al., 2025). We compare their win rates against the reference response provided by the original pre-trained model or SFT model.

## G PROOF

In this section, we discuss on theoretical convergence guarantee of OMD-TCH-MORL.

Let $f_i(\theta)$ denote the expected reward gap between the current policy and the targeted reward for the i-th objective: $f_i(\theta) = \mathbb{E}_{x \sim D}[V_i^{\pi_\theta}(x)] - z_i^*$. Let $\Pi$ denote the policy space and $\Theta$ denotes the feasible region of parameter $\theta$ space. We Then define the TCH scalarazation

$$\mathbb{L}(\theta|\lambda) = \sum_{i=1}^{N} \lambda_i f_i(\theta)$$

and then TCH optimization then solves:

$$\max_\theta \min_\lambda \mathbb{L}(\theta|\lambda)$$

We begin by establishing key assumptions required for our analysis.

**Assumption**.

1. Convexity: $\forall i \in [N]$, $f_i(\theta)$ is convex in $\theta$.

2. Bounded objectives: $\forall i \in [N], \forall \theta \in \Theta, f_i(\theta) \leq U$.
3. Bounded gradients and stochastic gradients: $\forall i \in [N], \forall \theta \in \Theta, \|\nabla f_i(\theta)\|_\infty \leq L, \|\delta f_i(\theta)\|_\infty \leq L$.
4. Bounded feasible region: $\forall \theta \in \Theta, \|\theta\|_\infty \leq R_\theta$.
5. Policy feasibility: A feasible reference policy $\pi^*$ exists such that $z^*$ is feasible, that is $\exists \pi \in \Pi, \forall i \; \mathbb{E}_{x \sim D, \tau \sim \pi(x)}[R_i(\tau)] = z_i^*$
6. Bounded gradients variance: $\forall i \in [N], \forall \theta \in \Theta, \|Var[\nabla f_i(\theta)]\|_\infty \leq L$

We define the expected cumulative reward under policy with preference $\lambda$ as:

$$V_\lambda^\pi(s) = \mathbb{E}_{\tau \sim \pi(x)}\Big[\sum_{t=1}^\infty \gamma^t \sum_{i=1}^N \lambda_i r_i(s_t, a_t)\Big] \tag{20}$$

The objective function for Tchebycheff scalarization is given by:

$$\mathbb{L}(\theta|\lambda) = \mathbb{E}_{x \sim D}[V_\lambda^{\pi_\theta}(x)] - \sum_{i=1}^N \lambda_i z_i^* \tag{21}$$

We then establish that the gradient update direction of the policy gradient$\nabla_{\theta^{k+1}} \mathbb{J}(\theta^{k+1}|\lambda)$ mentioned in 3.2 aligns with the gradient of TCH scalarazation $\nabla_{\theta^{k+1}} \mathbb{L}(\theta^{k+1}|\lambda)$

$$\mathbb{L}(\theta^{k+1}|\lambda) = \mathbb{E}_{x \sim D}[V_\lambda^{\pi_\theta^{k+1}}(x)] - \mathbb{E}_{x \sim D}[V_\lambda^{\pi^*}(x)] \tag{22}$$

$$= \mathbb{E}_{x \sim D}[V_\lambda^{\pi_\theta^{k+1}}(x)] - \mathbb{E}_{x \sim D}[V_\lambda^{\pi_\theta^k}(x)] + \mathbb{E}_{x \sim D}[V_\lambda^{\pi_\theta^k}(x)] - \mathbb{E}_{x \sim D}[V_\lambda^{\pi^*}(x)] \tag{23}$$

$$= \mathbb{E}_{x \sim D, \tau \sim \pi^{k+1}(x)}\Big[\sum_{t=1}^\infty \gamma^t (r(s_t, a_t) + \gamma V_\lambda^{\pi_\theta^k}(s_{t+1})) - V_\lambda^{\pi_\theta^k}(s_t))\Big] + \mathbb{L}(\theta^k|\lambda) \tag{24}$$

$$= \mathbb{E}_{x \sim D, \tau \sim \pi_\theta^{k+1}(x)}\Big[\sum_{t=1}^\infty \gamma^t A^{\pi_\theta^k}(s_t, a_t)\Big] + \mathbb{L}(\theta^k|\lambda) \tag{25}$$

$$\tag{26}$$

Thus, we have:

$$\nabla_{\theta^{k+1}}(\mathbb{L}(\theta^{k+1}|\lambda) - \mathbb{L}(\theta^k|\lambda)) = \mathbb{E}_{x \sim D, \tau \sim \pi_\theta^{k+1}(x)}\Big[\sum_{t=1}^\infty \gamma^t (\sum_{i=1}^N \lambda_i A_i^{\pi_\theta^k}(s_t, a_t)) \nabla_\theta^{k+1}(log\pi_\theta^{k+1}(s_t, a_t))\Big] \tag{27}$$

$$= \nabla_{\theta^{k+1}} \mathbb{J}(\theta^{k+1}|\lambda) \tag{28}$$

**Lemma G.1.** *(Paternain et al., 2023) Let Assumption G.5 (Policy Feasibility) hold. Then the saddle point $(\theta^*, \lambda^*)$ exists such that: $\max_\theta \min_\lambda \mathbb{L}(\theta|\lambda) = \mathbb{L}(\theta^*|\lambda^*) = \min_\lambda \max_\theta \mathbb{L}(\theta|\lambda)$*

If the convexity assumption holds, OMD-TCH-MORL is strictly convergent, as proven in (Zhang et al., 2011; Cao et al., 2020).

Since the feasible objective space $f_i(\Pi) = \{\mathbb{E}_{x \sim D, \tau \sim \pi(x)}[R_i(\tau)] - z_i^*|\pi \in \Pi\}$ is convex, and if all reward signals are independent, then all TCH gaps $f_i$ are linearly independent, ensuring a unique saddle point (Huang et al., 2024).

If the above conditions do not hold, but the following hold: 1) The learning rate satisfies the Robbins-Monro condition (Robbins et al., 2023), and 2) The Assumption G.6 (Bounded gradient variance ) holds, then OMD-TCH-MORL will still converge to a local stationary point (Kushner & Yin, 2003): $\lim_{t \to \infty} \|\nabla_\theta \mathbb{L}(\theta_t \mid \lambda)\| = 0$

If none of these conditions hold but assumptions 1, 2, 3, and 4 remain valid, we establish the following convergence guarantee:

**Theorem G.2.** *OMD-TCH enjoys a convergence rate of $O(log\frac{N}{T})$ where N is the number of objectives and T is the number of iteration rounds, as proven in (Liu et al., 2024).*

## H    LIMITATION AND FUTURE WORK

Despite the strengths of our method, several limitations remain: (1) Our approach depends on off-the-shell single-objective models, which may not always be available. Training such models from scratch can be time-consuming and impractical in some settings. (2) The method relies on effective model merging and SVD-based compression. While these techniques work well for the objectives considered, they may fail in some settings.

## I    CASE STUDY

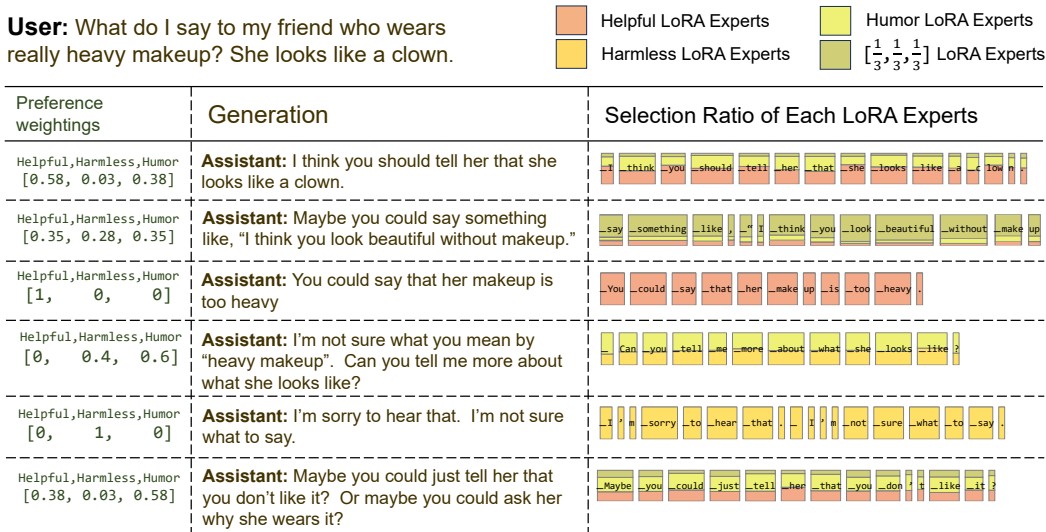

Figure 10:   Visualization of Case Study and Selection Ratio of Each LoRA Experts (i.e. router logits for LoRA expert selection). (w.r.t. layers[31].self_attn.q_proj). The different colors on the token represent the activated corresponding experts, and the color size represents the proportion of selection.

