# OpenReview forum: "Multi-objective Large Language Model Alignment with Hierarchical Experts"
_ICLR.cc/2026/Conference — ICLR 2026 Poster_

### Official Review · Reviewer_tavC · 2025-10-29

**Soundness:** 2
**Presentation:** 2
**Contribution:** 2
**Rating:** 4
**Confidence:** 3

**Summary:**

This paper proposes HoE, a hierarchical mixture-of-experts framework for multi-objective alignment of large language models, enabling them to adapt to diverse and conflicting human preferences without costly retraining. HoE integrates LoRA experts, router experts, and preference routing to efficiently cover the Pareto frontier, providing scalable and fine-grained control over model behaviour.

**Strengths:**

The paper propose a novel alignment approach named HOE. There are some strengths:
* **Methodology**: This paper introduces a hierarchical expert-model framework (HOE) to handle multi-objective alignment, and incorporates Pareto-optimality concepts to provide a theoretical grounding for the approach.
* **Scalable and extensible**: leverages model fusion techniques and a lightweight routing module to enable efficient training with lower resource costs.
* **Experiments**: The evaluation covers multiple datasets and baseline methods, yielding a comprehensive analysis of the results.

**Weaknesses:**

However, there are some weakness of this paper:
* **Methodology**: multi-objective LoRA experts are expected to learn different preferences, and the experimental results also validate it. However, the design of multi-objective router expert seems to be redundant. The ablation study only discusses a single router and the role and necessity of a multi-expert router are not clearly demonstrated.
* **Reproducibility**: The results appear to rely on the pretrained model, yet the manuscript does not specify how the pre-trained model was chosen. The method mainly fine-tunes the routing layer while freezing LoRA parameters, but there is no detail on where the LoRA parameters come from or whether they need to be trained. Taken together, these points suggest certain reproducibility gaps in the code and experimental setup.
* **Evaluation**: The evaluation in this paper primarily relies on reward-model scores, which may limit the ability to capture objective, user-centred aspects of quality and user needs. It would be more persuasive if the authors could explain the rationale behind the chosen metrics.

**Questions:**

* Q1: The HOE integrates the PPO paradigm to optimise models. I wondering that if the reward model in PPO is the same with the reward model in evaluation?
* Q2: In Line 1233, there may be citation error "(??)". Please correct it.
* Q3: The method mainly fine-tunes the routing layer while freezing LoRA parameters, but there is no detail on where the LoRA parameters come from or whether they need to be trained.

---

> ### Author Response · Authors · 2025-11-21
> **Rebuttal - Part I**
>
> > ###  Methodology: ... The ablation study only discusses a single router and the role and necessity of a multi-expert router are not clearly demonstrated.
>
> Thank you for raising this important point. To directly clarify this, we have conducted **substantial new ablation experiments for multi-router setting** and **updated Appendix B.3** with the full results.
>
> We isolate the roles of each router experts by incrementally removing or combining them to observe their effect on the Pareto frontier. We summarize key configurations and performance tables below (full details, and PF curves are now in Appendix B.3):
>
> ---
>
> ##  1. Ablation  Experiments for 3-Objective
> We tested the following five multi-router settings on Llama 3.1–8B, evaluated on HelpSteer2 and Psoups under Helpful–Harmless–Humor. Performance tables below:
> | Setting | [1,0,0]| [0.5,0.5,0]  |  [0,1,0]|  [0,0.5,0.5] |  [0,0,1]|  [0.5,0,0.5] |  [0.33,0.33,0.33]|
> | ---------|---------|---------|---------|---------|---------|--------|---------|
> | Base                            | 1.15 | 0.91 | 0.85 | -0.07 | -0.83 | 0.1  | 0.32 |
> | 4LoRA\&1router             | 3.05 | 1.02 | 2.25 | 1.79 | 2.03 | 1.78 | 1.03 |
> | 3LoRA\&1router             |  -   | 0.97 |   -  | 1.58 |  -   | 1.53 | 1.03 |
> | 3LoRA\&4router            |  -   | 0.99 |   -  | 1.62 |  -   | 1.65 | 1.03 |
> | 4LoRA                    |  -   | 1.01 |   -  | 1.70 |  -   | 1.69 | 1.06 |
> | 4LoRA\&3router             |  -   | 1.05 |   -  | 1.86 |  -   | 1.91 | 1.15 |
>
>
> | Setting | [1,0,0]| [0.5,0.5,0]  |  [0,1,0]|  [0,0.5,0.5] |  [0,0,1]|  [0.5,0,0.5] |  [0.33,0.33,0.33]|
> --- | --- | --- | --- | --- | --- | --- | ---
> | Base                           | 0.72 | 0.63 | 0.55 | -0.04 | -0.51 | 0.09 | 0.17 |
> | 4LoRA\&1router            | 2.36 | 0.66 | 1.5 | 1.58 | 1.96 | 1.63 | 1.01 |
> | 3LoRA\&1router           |  -   | 0.64 |  -  | 1.26 |  -   | 1.27 | 0.79 |
> | 3LoRA\&4router            |  -   | 0.64 |  -  | 1.37 |  -   | 1.42 | 0.92 |
> | 4LoRA                   |  -   | 0.65 |  -  | 1.48 |  -   | 1.56 | 1.04 |
> | 4LoRA\&3router           |  -   | 0.69 |  -  | 1.64 |  -   | 1.71 | 1.12 |
>
> ---
>
>
>
> ##   2. Ablation  Experiments for 5-Objective
> We tested the following three settings on Llama 3.1–8B evaluated on HelpSteer. Performance tables below:
> | Setting | Preference1  |  Preference2|  Preference3 |  Preference4|  Preference5 |  Preference6|
> | ---------|---------|---------|---------|---------|---------|--------|
>  |6LoRA\&1router        |     62.1   |    59.7     |  58.8   |   63.0     |  63.3   |   63.8   |
> |6LoRA\&3router        |   62.9    |    60.5     |  59.2   |   63.1     |  63.4   |   63.7     |
> |6LoRA\&5router        |   62.9    |    60.5     |  59.2   |   63.5     |  63.9   |   64.0     |
>
> ---
>
> ##  Key Findings (consistent across 3-obj and 5-obj):
>
> 1.	**Router experts consistently improve PF coverage.**
> Each router introduces new preference directions that increase the density and smoothness of the Pareto frontier.
> 2.	**Gains are monotonic and reproducible.**
> Both in aggregated scores and PF geometry, more routers → strictly better approximation.
> 3.	**Router experts provide benefits that LoRA expansion alone cannot replicate.**
> Even with additional multi-objective LoRA experts, removing router experts significantly degrades PF coverage.
>
> These results clearly show that router experts are not redundant—and equally, LoRA experts cannot fully substitute router experts. Their combination delivers the best performance–efficiency trade-off.
>
> ---
>
> **Why the ablation in the original paper used only one router?**
>
> For 2-objective alignment, a single router already performs extremely well because the user preference set is a line segment. Therefore, although additional routers can still help, the marginal benefit is small enough that we used the simplest setting in the ablation.
> We clarify this explicitly in the revision.

---

> ### Author Response · Authors · 2025-11-21
> **Rebuttal - Part II**
>
> >  ### Methodology:  the design of multi-objective router expert seems to be redundant.
>
> Beyond the new experimental results provided in Appendix B.3, *we now would also like to clarify the design rationale behind introducing router experts in HoE*. Router experts are not redundant; rather, they serve as a complementary and lightweight mechanism that strengthens multi-objective control in three important ways:
>
> ### 1. **Comparable Performance to LoRA Experts**:
>
> Our **ablation study (Figure.5 Left) and updated experiments (Appendix B.3)**  show that *router experts consistently deliver gains comparable to adding additional multi-objective LoRA experts*, especially in regions where the benefit of increasing the number of LoRA experts begins to show diminishing returns.
> As illustrated in Figure 5 (Left) of the main paper, adding one router expert to three single-objective LoRA experts achieves performance similar to adding a fourth multi-objective LoRA expert.
> Thus, router experts provide an efficient and scalable alternative to simply stacking more multi-objective LoRA modules.
>
> ### 2. **Minimal Parameter Overhead**:
>
> While an individual LoRA expert is small, adding many multi-objective LoRA experts eventually incurs non-negligible parameter and memory overhead. In contrast, router experts introduce **virtually negligible parameter cost**, yet contribute significantly to PF coverage and controllability. This extremely low overhead allows HoE to keep the method lightweight and avoid accumulating dozens of LoRA adapters. Although technologies like S-LoRA support efficient inference with hundreds of LoRA modules, we wanted to demonstrate that HoE can perform well with minimal extra cost, keeping the method lightweight and broadly accessible.
>
> ### 3. **Fine-Grained, Module-Wise Preference Control (a capability LoRA experts alone cannot provide)**:
>
> Router experts also offer a qualitatively different capability: **module-wise, input-adaptive, fine-grained routing** across experts. Unlike LoRA experts—whose combination is static and tied to fixed preference vectors—router experts dynamically determine expert activation at each layer and (in practice) often at the token level.
>
> **Our case study illustrates this vividly (Fig. 6)**. Under a intentionally adversarial user prompt and mixed preference vector [0.35, 0.28, 0.35], the model performs:
> -  Early tokens ``Maybe you could say something like,''→ dominated by the Helpful expert
> -  Later tokens ``I think you look beautiful without makeup.''→ increasing influence from Harmless + Humor experts
>
> *This token-level trade-off among experts is unique to HoE and cannot be achieved by LoRA merging alone*. It offers interpretable and flexible preference control that goes beyond existing MOA approaches.

---

> ### Author Response · Authors · 2025-11-21
> **Rebuttal - Part III**
>
> > ### Evaluation: The evaluation in this paper primarily relies on reward-model scores, which may limit the ability to capture objective, user-centred aspects of quality and user needs. It would be more persuasive if the authors could explain the rationale behind the chosen metrics.
>
> Thank you for raising this concern. We acknowledge that our evaluation primarily relies on reward-model (RM) scores, and we agree that RMs—while widely used—are not perfect. To address potential bias and to justify our metric choices, we provide several clarifications and additional evidence below.
>
>
> ### 1. **We carefully selected open-source, community-recognized RMs with low known bias**
>
> All reward models used in this paper are publicly available, widely adopted in prior alignment work, and recognized for having comparatively low bias. We explicitly **list them in Appendix F.2**, covering multiple objectives:
> - Helpful / Harmless: RMs from Rewarded-Soups (RS)
> - Humor: RM trained from a dedicated humor preference dataset
> -	Helpfulness / Correctness / Coherence / Complexity / Verbosity: multi-dimensional RMs from DPA
> -	Reward / Cost: RMs from Safe-RLHF
> -	Faithfulness / Summary / DeBERTa-based evaluators: RMs trained by OpenAssistant and related communities
> These RMs originate from different institutions, datasets, and training pipelines, ensuring diversity across preference axes.
>
>
> ### 2. **We perform extensive cross-evaluation using non-RM metrics (Appendix F.5)**
>
> To ensure our findings are not tied to any single RM family, we include several independent evaluation signals:
> - **Math reasoning**: PASS@1 on GSM8K + number of CoT tokens
> - **Code generation**: PASS@1 on HumanEval
> - **Over-refusal benchmark**:
>   - Safety score: probability of resisting harmful jailbreak prompts
>    - Helpfulness score: success rate on benign-but-misleading prompts
> - **Assistant-style dialogue**: GPT-4 comparative win rate against reference responses
> These metrics cover *accuracy, robustness, safety, and user-perceived quality*, offering **complementary perspectives beyond RM** scores.
>
>
> ### 3. **Commitment to add additional qualitative case studies demonstrating human-centered alignment**
> We will further include extensive case studies in Appendix I as soon as possible, illustrating how HoE adapts to user preferences in realistic scenarios. These qualitative results will show interpretable preference trade-offs and human-aligned behavior without evidence of reward hacking. We hope these examples can complement quantitative scores and directly address the reviewer’s concern about user-centered evaluation.
>
> ### 4. **Reward models are the community-accepted proxy for human preference, especially in multi-objective alignment**
> Our evaluation **follows the standard practice** used by MODPO, RS, RiC, DPA, MOD, Args, PAD, GenARM, RARM, and many other alignment and RLHF papers.
> While RMs inevitably contain some bias, MOA (multi-objective alignment) **specifically focuses on modeling and controlling trade-offs between heterogeneous preferences**. In this context, RM biases can be viewed as expressive preference directions rather than noise. For example, the three summary-evaluation RMs we use emphasize different aspects—relevance, fidelity, verbosity, or holistic quality—due to differences in their base models and training data. This diversity provides multiple human-centered signals and allows HoE to model fine-grained trade-offs across these dimensions.
>
> ### 5. **RMs are the only scalable proxy for large-scale preference evaluation**
> Human evaluation or rule-based metrics (ROUGE, BERTScore) cannot capture fine-grained preference differences or scale to thousands of dense sampling across a high-dimensional PF.
> In contrast, reward models enable dense, continuous evaluation across many preference directions  and are widely employed in GPT, Llama, DeepSeek, and Anthropic’s constitutional AI pipelines
> Given that large-scale Pareto alignment requires evaluating thousands of model responses at different preference vectors, RMs are **the only scalable and standardized solution.**
>
> ---
>
> Thank you for raising this concern. We hope this clarifies the rationale behind our evaluation methodology and reinforces the validity of our results.

---

> ### Author Response · Authors · 2025-11-21
> **Rebuttal - Part Ⅳ**
>
> > ### Reproducibility: The results appear to rely on the pretrained model, yet the manuscript does not specify how the pre-trained model was chosen. The method mainly fine-tunes the routing layer while freezing LoRA parameters, but there is no detail on where the LoRA parameters come from or whether they need to be trained. Taken together, these points suggest certain reproducibility gaps in the code and experimental setup.
>
> Thank you very much for raising this point. The comment touches on two closely related questions:
>  - (1) Where do the LoRA parameters come from?
>  - (2) Do LoRA parameters need to be trained?
>
> While both questions were **discussed in our original Appendix (Sections F.3 and B.2) and Line 241**, we agree that **the explanation could be made clearer**. For completeness and to avoid any ambiguity, we restate and clarify the answers here.
>
> ---
>
> ### 1. **Where do the LoRA parameters come from?**
>
> Our LoRA experts are obtained through two standard and fully reproducible sources, depending on availability:
>
> **(a) Extracting LoRA experts from open-source, off-the-shelf preference models**
>
> Whenever possible, we directly extract LoRA experts from publicly available preference-oriented models.
> As detailed in **Appendix F.3**, we utilize:
> - Math-specific experts: *three open-source MathLlama variants*
> - Code-specific experts: *two CodeLlama variants*
> - Helpfulness / Harmlessness experts: *three open-source fine-tuned Llama models trained on safety and alignment datasets*
>
> HoE is intentionally designed to work with widely available pre-trained models, reducing the user’s training burden and allowing the community to fully reproduce our expert extraction pipeline.
>
>
> **(b) Training single-objective experts from scratch when no open-source models exist**
>
> In cases where we could not locate a suitable open-source model for a specific objective, we **train single-objective models from scratch** using the RiC (Reward-in-Context) training pipeline.
> We report the exact cost of training three such single-objective models in **Appendix B.2**, ensuring transparency.
>
>
> **(c) For maximal reproducibility, we release all LoRA experts used in the Llama-3.1 experiments**
>
> These LoRA adapters—extracted using our pipeline—are already uploaded to our **public GitHub repository**.
> Each expert functions exactly like a standard LoRA adapter and can be loaded independently, allowing anyone to verify their performance. This guarantees full reproducibility of both our expert construction and downstream HoE results.
>
> ---
>
> ### 2. **Do LoRA parameters need to be trained?**
>
> As detailed in  Line 241, **HoE freezes all LoRA parameters and fine-tunes only the routing experts.**
> Thus, LoRA parameters do not need to be trained during HoE’s preference optimization stage.
> Below we provide our motivation for freezing LoRA parameters:
>
> **(a) Training efficiency and the philosophy of decomposition**
>
> Fine-tuning many LoRA experts jointly across all preferences would:
> - cause significant computational overhead, since all experts must be updated simultaneously
> -  contradict the core motivation of decomposition, where each expert should encode a stable, isolated preference
> -  require re-adjusting previously trained experts whenever the router is updated
> -  hurt interpretability, since shared updates entangle the experts and obscure their preference specialization
>
> In contrast, HoE aims for **lightweight training and true plug-and-play modularity**, where new preference capabilities can be added or combined without retraining existing experts.
>
> **(b) Avoiding severe gradient conflicts during multi-objective training**
>
> Using Helpful and Harmless for example, the policy gradients for these two objectives exhibit strong negative correlations (up to -0.7). This causes severe gradient interference under dynamically sampled preferences, where a single model trained jointly on multiple objectives faces opposing updates, leading to instability in the RL training process.  HoE avoids these issues by **freezing each LoRA expert**, ensuring that **each one remains a stable encoding of a single-preference**.
> The resulting training stability improvement is visualized at the following anonymous link:
> (https://anonymous.4open.science/r/training_process-005/training_process.pdf)
>
> ---
>
> Thank you very much for raising this point. We hope this addresses the reviewer’s concerns regarding reproducibility and clarifies our experimental setup.

---

> ### Author Response · Authors · 2025-11-21
> **Rebuttal - Part V**
>
> > Q1: The HOE integrates the PPO paradigm to optimize models. I am wondering whether the reward model used in PPO is the same as the reward model used for evaluation?
>
> Thank you for the question. Yes — the reward model used during PPO training is exactly the same as the one used during evaluation. We have updated the text to make this explicit.
>
> > Q2: In Line 1233, there may be a citation error (“(??)”). Please correct it.
>
> We appreciate the reviewer for pointing this out. The citation has now been corrected in the revised manuscript.
>
> > Q3: The method mainly fine-tunes the routing layer while freezing LoRA parameters, but there is no detail on where the LoRA parameters come from or whether they need to be trained.
>
> Thank you for highlighting this important point. As clarified in our response to Weakness 2 in Rebuttal - Part IV, we have now added a detailed explanation addressing the origin and training status of the LoRA experts. These clarifications are also reflected in the revised Appendix.
>
> ---
>
> ## Once again, we sincerely appreciate the reviewer’s time and thoughtful feedback. Your comments have helped us significantly strengthen the clarity and completeness of our manuscript. If any further concerns remain, we would be more than happy to address them.

---

### Official Review · Reviewer_V6P4 · 2025-10-30

**Soundness:** 3
**Presentation:** 2
**Contribution:** 3
**Rating:** 6
**Confidence:** 3

**Summary:**

The paper tackles the important and challenging problem of multi-objective alignment for LLMs. The core idea of using a hierarchical, decomposition-based MoE framework (HOE) is novel and parameter-efficient. The paper's primary strength lies in its comprehensive and very strong empirical results, demonstrating state-of-the-art performance across numerous benchmarks by dominating 15 baselines.

**Strengths:**

1. The core architectural idea of a hierarchical Mixture-of-Experts for MOA, inspired by decomposition methods, is novel.

2. HOE achieves state-of-the-art performance, consistently dominating the Pareto frontiers of 15 competitive baselines (including RS, MOD, and RiC) in 2-objective settings. This is a very strong empirical contribution.

3. The paper features high-quality ablation studies that provide clear insights into the model's components.

**Weaknesses:**

1. The paper's central claim is the achievement of "optimal Pareto frontiers" and "superior Pareto-optimal results". However, the paper provides no evidence that the proposed HOE method actually converges to the true, global Pareto optimal frontier. The theoretical analysis in Appendix G relies on strong assumptions, such as the convexity of the objective functions (Assumption G.1), which are well-known to not hold in the non-convex landscape of LLM optimization. While the use of Tchebycheff (TCH) scalarization is appropriate (as it can find non-convex frontiers), it does not guarantee that the frontier it finds is the optimal one. Therefore, all of the paper's claims of "optimality" are purely empirical and lack the foundational theoretical guarantees that the term "Pareto optimal" implies.

2. The framework's design is confusing, as it introduces two distinct types of multi-objective components: "Multi-Objective LoRA Experts"  and "Router Experts". Both components appear to serve the exact same purpose: covering intermediate points on the Pareto frontier. The paper fails to provide a clear justification for why both are necessary.

3. This paper lacks of criticial details. A core component, the Merge function used to create multi-objective experts ($\tau_{\lambda}=Merge(...) $), is never defined in the paper.  The "task-SVD" compression process is also vague. The example in Appendix E.1  suggests it may involve manual, per-objective hyperparameter tuning, which would severely undermine the "plug-and-play" and "lightweight" claims. Furthermore, the mathematical formulation for the router optimization is inconsistent between the main text and the appendix, confusing any attempt at re-implementation.

4. There are several issues for the Proof in App. G as well.

First, the proof's strongest claims of convergence rely entirely on Assumption G.1 (Convexity). The parameters $\theta$ being optimized belong to the Router Experts, which are neural network layers within a Transformer. The optimization landscape for LLMs and deep neural networks is well-known to be highly non-convex.

Second, even if local stationary point convergence is established, a local stationary point is not, by any means, equivalent to a Pareto optimal solution.

Third, Lines 1480-1494 are very confusing. For example: “If none of these conditions hold but assumptions 1, 2, 3, and 4 remain valid..."

5. Please clarify some mathematical formulations. The main text in Section 3.2 (Eq. (6)-(8)) introduces an Online Mirror Descent (OMD) method for the Tchebycheff (TCH) objective . However, Appendix E.3 (Eq. (12)-(18)) shows a different-looking formulation that is supposedly based on the same TCH and OMD principles. The relationship between Eq. (8)  and Eq. (18)  is unclear, even though both appear to describe the same PPO gradient

**Questions:**

Please see the Weaknesses above.

---

> ### Author Response · Authors · 2025-11-21
> **Rebuttal - Part I**
>
> > ### The framework's design is confusing, as it introduces two distinct types of multi-objective components: "Multi-Objective LoRA Experts" and "Router Experts". Both components appear to serve the exact same purpose: covering intermediate points on the Pareto frontier. The paper fails to provide a clear justification for why both are necessary.
>
> Thank you for raising this thoughtful concern. We agree that both *Multi-Objective LoRA Experts* and *Router Experts* contribute to covering intermediate regions of the Pareto frontier. However, similar purpose does not imply redundancy. These components play distinct, complementary roles, and our design choices are grounded in both **empirical evidence** and **principled motivation** for scalable multi-objective alignment.
>
> ---
>
> ## 1. Strong Empirical Evidence: Both components are indispensable
>
> We added **substantial new ablation experiments (now in Appendix B.3)** specifically studying multi-router  settings. We isolate the roles of LoRA experts and router experts by incrementally adding or removing experts and measuring their effects on the Pareto frontier (PF).
>
> 3-objective settings tested:  *(1) 3LoRA & 1 Router; (2) 4LoRA; (3) 4LoRA & 1 Router; (4) 3LoRA & 3 Router; (5) 4LoRA & 3 Router.*
>
> 5-objective settings tested: *(1) 6LoRA & 1 Router; (2)  6LoRA & 3 Router; (3) 4LoRA & 5 Router;*
>
> ###  Key Findings (consistent across 3-obj and 5-obj):
>
> 1.	**Router experts consistently improve PF coverage.**
> Each router introduces new preference directions that increase the density and smoothness of the Pareto frontier.
> 2.	**Gains are monotonic and reproducible.**
> Both in aggregated scores and PF geometry, more routers → strictly better approximation.
> 3.	**Router experts provide benefits that LoRA expansion alone cannot replicate.**
> They play a central role whenever M > 2 objectives, and their effect cannot be replicated by only adding LoRA experts.
>
> ---
>
> ## 2. Clarifying HoE’s Motivation and the Distinct Roles of Each Component
> ### 2.1 Why Multi-Objective LoRA Experts alone are not enough
> Increasing the number of Multi-Objective LoRA experts does improve PF coverage, but at a cost:
> - Each LoRA adapter contributes **non-trivial parameter overhead**, and many such adapters quickly become prohibitive.
> - Maintaining dozens of LoRA adapters is undesirable for deployment, storage, and model distribution.
> - LoRA experts are static—their influence is fixed once trained.
>
> Thus, relying solely on LoRA experts is not scalable.
>
> ---
>
> ###  2.2 Why Router Experts are necessary?
> Router experts provide:
>
> **(a) Comparable performance to additional LoRA experts**
> As shown in **Figure 5 (Left)** and new ablation experiments in **Appendix B.3**:
> Adding one router provides gains similar to adding one additional MO-LoRA expert, especially when LoRA benefits begin saturating.
>
> **(b) Negligible parameter overhead**
> A router is orders of magnitude smaller than a LoRA adapter.
> Thus, routers deliver “LoRA-like improvements” with almost no cost.
>
> **(c) Fine-grained, dynamic, token-level routing**
> Routers enable input-adaptive, layer-wise, and often token-level expert combination—capabilities fundamentally impossible for static LoRA merging.
> Our case study (Fig. 6) visually demonstrates this:
>  - Early tokens → dominated by Helpful
>  - Later tokens → smoothly transition to Harmless + Humor
>
> All within a single response, enabling dynamic preference steering.
> This type of context- and token-adaptive trade-off is unique to HoE and cannot be obtained with LoRA modules alone.
>
> ---
>
> ### 2.3 Why Router Experts alone are not enough
>
> Router experts, although lightweight, do not increase representation capacity. They recombine existing LoRA experts—**they cannot invent brand new preference-specific behaviors.**
>
> If we remove all Multi-Objective LoRA experts (e.g., *3LoRA & 4Router* settings in 3-obj ablation experiment ), routers gain very limited expressive power  (as shown in **Figure 5 (Left)** and updated **Table 4**). Routers cannot compensate for missing multi-objective behaviors and PF coverage collapses because single-objective LoRAs only cover endpoints.  Therefore, routers alone cannot approximate the PF sufficiently.

---

> ### Author Response · Authors · 2025-11-21
> **Rebuttal - Part II**
>
> > ### the paper provides no evidence that the proposed HOE method actually converges to the true, global Pareto optimal frontier. it does not guarantee that the frontier it finds is the optimal one. Therefore, all of the paper's claims of "optimality" are purely empirical and lack the foundational theoretical guarantees that the term "Pareto optimal" implies.
>
> Thank you for pointing out this important issue. **We agree that our original use of the term “optimal” may have been inappropriate** and could mislead readers into assuming theoretical guarantees that our method does not claim to provide. We sincerely apologize for this confusion. In our revision, **we have removed all expressions implying global Pareto optimality:**
> - For empirical results, we now refer strictly to state-of-the-art performance.
> - For theoretical descriptions, we replace “optimal” with “near-optimal” and clearly articulate the limitations of our guarantees.
>
> ---
>
> ### 1. Original intent of “optimal” in the manuscript (now fully revised)
>
> The term “optimal” in the original submission was meant to convey two empirical observations.
> -  **Empirical superiority**:
> Across all experiments, HoE consistently outperforms prior baselines along multiple preference weightings. In this sense, “optimal” was intended to mean the best among existing methods, not the true theoretical optimum.
> - **Idealized limit case**:
> In a conceptual sense, if each single-objective expert were itself optimal, then their combination would—under ideal conditions—approximate the true PF. This was meant as a motivation, not a formal theorem.
>
> To avoid misunderstanding, we have removed such phrasing and revised all instances to reflect the empirical nature of our claims.
>
> ---
>
> ### 2. Empirical support for “near-optimality” (not theoretical optimality)
>
> We highlight that HoE’s empirical performance strongly supports its ability to approximate high-quality PFs.
> Thus, our empirical use of “optimal” referred only to state-of-the-art results. We now phrase this precisely as such.
>
> For example, in Lines 302–305 of the original manuscript, we wrote that HoE “closely approaches MORLHF’s optimal results.” Here, we were using **MORLHF as an empirical performance upper bound**, since it directly optimizes the PF through costly multi-objective RLHF. We relied on this empirical framing, not a theoretical guarantee.
>
> ---
>
> ### 3. Theoretical motivation for “near-optimal” behavior (without claiming formal guarantees)
>
> Our method aims to reconstruct a high-quality approximation of the PF using a decomposition–composition strategy:
>
>  - *Decomposition*:
> Each LoRA expert specializes in a single objective or isolated preference direction.
> We assume (empirically supported) that each expert produces near-optimal behavior for its subproblem.
>
>  - *Composition*:
> When a new preference vector is presented at inference time, HoE performs a locally linear combination of neighboring experts via the preference routing strategy, similar to locally linear Rewarded Soups (RS).
>
> Under two mild empirical assumptions:
>  - each expert is near-optimal for its own weighting,
>  - local RS-style interpolation preserves convexity,
>
> then the composed solution lies on or near the true PF.
>
> ---
>
> ### We really appreciate the reviewer’s careful reading—this feedback helped us improve the precision and clarity of the manuscript.

---

> ### Author Response · Authors · 2025-11-21
> **Rebuttal - Part III**
>
> > ### This paper lacks of criticial details. the Merge function ......The "task-SVD" compression process
>
> Thank you for raising these important points. We fully agree with your point, and we have now **substantially expanded Appendix E.1** to clarify all components mentioned by the reviewer.
>
> The revised manuscript specifies that HoE uses: **PCB-Merging combined with DARE for merging** and HoE uses: **Activation-Aware SVD (A-SVD) for compression.** These steps, along with their mathematical forms and pseudocode, are now provided in Appendix E.1.
>
> We also test other compression alternatives:
> - Naïve SVD → high accuracy loss
> - A-SVD → best trade-off, used in all experiments
> - Delta-Come → highest fidelity but requires GPTQ training, so not used in the paper
>
> ---
>
> > ### The example in Appendix E.1 suggests it may involve manual, per-objective hyperparameter tuning, which would severely undermine the "plug-and-play" and "lightweight" claims.
>
>
> We understand this concern. However, we clarify that while HoE does require a small amount of per-objective hyperparameter tuning during the merging stage, the amount of tuning is **extremely small** and **negligible compared to any form of training.**
>
> **(a) Only 2–4 hyperparameters are ever tuned**
>
> For each LoRA merging step, the only tunable values are:
>
> 1.	Rescale coefficient (λ): grid search over 1.6–2.2 (7 values)
> 2.	SVD clamp threshold: grid search over 0.2–0.4 (3 values)
> 3.	Rank: typically fixed in advance (128–512), not tuned per objective
> 4.	Merging weightings: typically equal to expected preference weighting
>
> All other parameters are fixed:
> - PCBmerging.att_ratio = 0.9 (never tuned)
> - clamp.min_ratio, clamp.max_ratio= 0.01 (stable across all tasks)
> - merging normalization = tanh (works universally)
>
> Thus the total tuning space is extremely small.
>
> **(b) The total tuning time is negligible (< 1–2 hours), compared to 30–50hours of training**
>
> For example, merging three 8B models using 7 × 3 grid search on 4×A100-80GB takes:
> - **< 1 hour** for merging
> - **< 2 hour** for evaluation
>
> By contrast:
> - Single-objective RSFT training requires **10–13 hours per model**
> - PPO/RLHF training requires **30–45 hours per model**
>
> Thus even in the worst case, tuning time is <2% of training cost.
> This is why model-merging methods—PCBMerging, TiesMerging, DARE—**are widely considered “training-free” in the literature.**
>
> **(c) Plug-and-play property is preserved**
>
> All tuning occurs **before** HoE is created. Once the LoRA experts are created, they never need to be retrained.
>
> ---
>
> > ### the mathematical formulation for the router optimization is inconsistent between the main text and the appendix, confusing any attempt at re-implementation. The relationship between Eq. (8) and Eq. (18) is unclear, even though both appear to describe the same PPO gradient
>
> Thank you for pointing this out.  The discrepancy originates from **our attempt to compress notation in the main text due to space constraints.**
>
> Both Eq. (8) and Eq. (18) implement the **same optimization process**, but differ in their level of abstraction: Section 3.2 presents a high-level, compact form of the Tchebycheff (TCH) scalarization; Appendix E.3, by contrast, expands these expressions fully and combined Online Mirror Descent (OMD)
>
> To avoid further confusion, we have **rewritten Appendix E.3** to explicitly map each term in Eq. (8) to its corresponding term in Eq. (18), and we now clarify the connection step-by-step.  **We thank the reviewer for catching this and have improved the clarity accordingly.**
>
> ---
>
> > ### Several Concerns About the Proof in Appendix G
>
> Thank you very much for the reviewer’s careful reading of Appendix G and for raising these important concerns. We agree that several aspects of the original presentation may have not been communicated clearly, and the current wording could mislead readers.
>
> For example, regarding the Assumption G.1 (Convexity): $f_i(\pi_\theta) = E_{x\sim D}[V_i^{\pi_\theta}(x)] - z^{\ast}_i$
>
> Our intention was not to imply convexity over the parameter space $\Theta$. Instead, the analysis considers convexity with respect to the policy space $\Pi$,  where the scalarized objective $f_i(\pi_\theta)$ is convex for $\pi_\theta$ in $\Pi$
>
>
> We are grateful to the reviewer for pointing out such issues. We are committed to ensuring that the theoretical discussion is accurate, appropriately scoped, and free of misleading implications. We will continue refining the exposition to avoid similar inconsistencies.
>
> ---
>
> ## Once again, we sincerely appreciate the reviewer’s time and thoughtful feedback. Your comments have helped us significantly strengthen the clarity and completeness of our manuscript. If any further concerns remain, we would be more than happy to address them.

---

### Official Review · Reviewer_cfcq · 2025-11-04

**Soundness:** 3
**Presentation:** 3
**Contribution:** 3
**Rating:** 6
**Confidence:** 3

**Summary:**

The authors studied the problem of multi-objective alignment problem in LLMs. To address the problem, they decompose the alignment problem into a number of single-preference subproblems, each of which handled by specialized experts. They combined LoRA experts, router experts, and preference routing to address the problem in their hierarchical MOE framework.

**Strengths:**

The paper is well-written and easy to follow.
A number of different NLP tasks were taken to evaluate the performance of the proposed framework.
A number of datasets were used to conduct the experiments.

**Weaknesses:**

See the below questions.

**Questions:**

Motivations of why the authors combine LoRA experts, router experts, and preference for the multi-objective alignment problem are not convincing, as there are many methods/strategies to address such tasks.

Because the proposed framework is combined by LoRA experts, router experts, and preference, there are a large number of parameters applied in the framework, although the authors tried to reduce the size of the LLM by their lightweight, parameter-efficient, and plug-and-play approach. In fact, such combination may not be appropriate to address the multi-objective alignment ask due to the size of the LLMs after the combination.

Not sure if there are other ways to eliminate the need to train the proposed model beside the propose HoE?

Are there any additional baselines that were published in this year to be taken as baselines in the experiments?

---

> ### Author Response · Authors · 2025-11-21
> **Rebuttal - Part I**
>
> > ### Motivations of why the authors combine LoRA experts, router experts, and preference for the multi-objective alignment problem are not convincing, as there are many methods/strategies to address such tasks.
>
>
> Thank you for raising this thoughtful concern. We respectfully clarify that our architectural choices are grounded both in **empirical evidence** and in a **principled motivation** for scalable multi-objective alignment. Below, we address the reviewer’s point from three complementary angles.
>
> ---
>
> ## 1. Strong Empirical Evidence Supporting the Design Motivation
>
> HoE is motivated by two core ideas:
>
> (1) **Decomposing multi-objective alignment across local experts**, and
>
> (2) **Employing a hierarchical expert structure to balance performance and efficiency.**
>
> To validate these motivations, we conducted two groups of experiments.
>
> ---
>
> ## 1.1 Experiment Group I — Why combine LoRA experts and router experts?
>
> We added **substantial new ablation experiments (now in Appendix B.3)** specifically studying multi-router  settings. We isolate the roles of LoRA experts and router experts by incrementally adding or removing experts and measuring their effects on the Pareto frontier (PF).
>
> 3-objective settings tested:  (1) 3LoRA & 1 Router; (2) 4LoRA; (3) 4LoRA & 1 Router; (4) 3LoRA & 3 Router; (5) 4LoRA & 3 Router.
>
> 5-objective settings tested: (1) 6LoRA & 1 Router; (2)  6LoRA & 3 Router; (3) 4LoRA & 5 Router;
>
> ###  Key Findings (consistent across 3-obj and 5-obj):
>
> 1.	**Router experts consistently improve PF coverage.**
> Each router introduces new preference directions that increase the density and smoothness of the Pareto frontier.
> 2.	**Gains are monotonic and reproducible.**
> Both in aggregated scores and PF geometry, more routers → strictly better approximation.
> 3.	**Router experts provide benefits that LoRA expansion alone cannot replicate.**
> They play a central role whenever M > 2 objectives, and their effect cannot be replicated by only adding LoRA experts.
>
> These results empirically justify our motivation for combining LoRA experts with router experts.
>
> ---
>
> ### 1.2 Experiment Group II — Why decompose objectives across experts instead of joint training?
>
> Another core design choice is decomposing each preference into its own expert, instead of jointly finetuning all preferences within a single model. **We compare both training processes directly** (visualized in the anonymous link below): (https://anonymous.4open.science/r/training_process-005/training_process.pdf)
>
> ### Advantages of decomposition:
>
> **(a) Training efficiency & modularity**
>
> Jointly fine-tuning all LoRA experts across all preferences:
>  - incurs large computational overhead
>  - violates the decomposition principle (experts no longer encode stable preferences)
>  - forces re-adjustment of previously trained experts
>  - hurts interpretability due to parameter entanglement
>
> HoE, by contrast, is lightweight and plug-and-play: experts remain intact, and only the router needs training.
>
> **(b) Training stability & avoidance of gradient conflict**
>
> Preferences such as Helpful and Harmless exhibit strong negative gradient correlation (as large as –0.7).
> This causes severe gradient interference under jointly sampled preferences, destabilizing RL training.
> HoE resolves this by **freezing LoRA experts**, ensuring **each expert preserves a clean, preference-specific encoding**.
> This experiment confirms that decomposition is a core motivation—not an arbitrary design choice.
>
> ---

---

> ### Author Response · Authors · 2025-11-21
> **Rebuttal - Part II**
>
> ## 2. Clarifying and Reaffirming HoE’s Motivation
> Our method is motivated by a simple but powerful insight:
> The Pareto frontier can be reconstructed by combining localized preference-specific experts within a hierarchical MoE structure.
> HoE instantiates this idea: Each preference-specific behavior is captured by a lightweight expert, enabling smooth traversal of the PF and supports arbitrary user preferences, not just a fixed set.
>
> ---
>
> ## 2.1 Why router experts are necessary
>
> Router experts contribute three essential capabilities:
>
> 1.	**Comparable performance to LoRA experts**
> Adding one router often yields gains similar to adding an additional multi-objective LoRA expert (**Fig. 5, Left and  updated experiments in Appendix B.3**), especially when LoRA benefits saturate.
> 2.	**Negligible parameter overhead**
> Router experts add minimal memory cost and avoid the heavy overhead of maintaining many multi-objective LoRA adapters, as shown in our **cost analysis (Appendix B.2)**
> 3.	**Fine-grained, dynamic, module-wise expert routing**
> They enable token-level preference trade-offs that cannot be achieved via static LoRA merging alone.
> As shown in our **case study (Figure 6)**, router experts shift activation across Helpful / Harmless / Humor experts throughout a single response—offering interpretable and flexible control.
> Thus, combining LoRA experts, router experts, and preference vectors is a principled design, not an ad-hoc architecture.
>
>
> ---
>
>
> ## 3. The Unique Advantages of HoE in the MOA Landscape
> We appreciate the reviewer’s note that many strategies could be applied to multi-objective alignment.
> However, **HoE offers a unique combination of capabilities not found in existing methods.**
> A summarized version of Table 3 (“comparison checklist”) highlights HoE’s distinctive strengths:
>
> 1.	*Lightweight & Parameter-Efficient:*
> A single unified architecture replaces multiple separately trained models.
> 2.	*Predominantly Training-Free:*
> Only router experts require minimal training; LoRA experts are plug-and-play.
> 3.	*Minimal Inference Cost:*
> Unlike refinement- or multi-pass decoding methods (PAD, MOD, MetaAligner), HoE activates only a few small experts per query. (Table. 3)
> 4.	*Applicable to Multi-Task Learning:*
> Without any MTL-specific modifications, HoE achieves competitive MTL performance (Fig. 7).
> 5.	*Fully Pareto-Steerable:*
> Supports arbitrary user-defined preferences—not limited to a small set of preset objectives.
> 6.	*Scalable & Extensible:*
> New objectives can be added without retraining existing experts; simply extend the preference vector.
> 7.	*Prompt-Free & Robust:*
> Unlike prompt-based methods (PAD, DPA, MetaAligner), HoE handles abstract, non-verbalizable objectives (e.g., “DeBERTa score,” “Reward,” “Cost”) without prompt engineering.
>
> While each existing approach has strengths, they also have limitations (e.g., multi-pass decoding, prompt brittleness, retraining requirements). **HoE uniquely combines scalability, efficiency, modularity, and fine-grained preference control.**
>
> ---
> ---
> ---
>
> Thank you for raising this thoughtful concern. We hope this clarifies the principled motivation for combining LoRA experts, router experts, and preference vectors.

---

> ### Author Response · Authors · 2025-11-21
> **Rebuttal - Part III**
>
> > ### such combination may not be appropriate to address the multi-objective alignment ask due to the size of the LLMs after the combination.
>
> Thank you for raising this concern. We believe this issue arises from a misunderstanding: HoE does **not significantly increase model size**, and in fact, it is **one of the most parameter-efficient frameworks among all MOA baselines**. We clarify this with explicit evidence from our cost analysis.
>
> ---
>
> ## 1. Cost analysis in Appendix B.2 shows HoE is highly parameter-efficient
>
> As detailed in **Appendix B.2 (Cost Analysis)**, Table 3 provides a comprehensive comparison of inference-time parameter sizes across baselines. The results show:
> - HoE has **the second-lowest parameter footprint**, surpassed only by RS and MOD.
> - HoE is substantially smaller than multi-model or multi-pass approaches such as MetaAligner, PAD, and RLHF.
>
> The reviewer’s impression that HoE introduces “a large number of parameters” is not accurate—our quantitative analysis demonstrates the opposite.
>
> ---
>
> ##  2. HoE’s parameter advantage becomes even more pronounced compared to full-parameter fine-tuning
>
> Table 3 reports LoRA-based fine-tuning settings. If we compare HoE under **full fine-tuning settings**, the difference becomes even larger:
>
> In real deployment scenarios—e.g., optimizing the trade-off between reasoning length and accuracy—this typically requires maintaining three separate LLMs:
> - a long-CoT reasoning model
> - a short non-reasoning model
> - a distilled short-CoT model
>
> → totaling **3× the parameter size**.
>
> In contrast:
> - HoE uses only one shared backbone,
> - plus a few small LoRA experts (e.g., rank 512–1024 for a 7B model),
> - resulting in **< 2× total parameters size**,
>
> This shared-backbone design is precisely why HoE is more scalable and practical for deployment than traditional multi-model approaches.
> Figure 3 (upper left subfigure) shows that HoE reconstructs the Pareto frontier effectively in this scenarios.
>
> ---
>
> ## 3. HoE has no inference-time disadvantage—and is faster than many baselines
>
> While the size of HoE indeed grow with the number of experts, **inference cost remains low**. As cited in **Appendix B.2**, we adopt S-LoRA as inference engine. S-LoRA allow 100+ experts to be served concurrently on a single GPU，and achieves 2–3× speedup over vLLM’s LoRA serving when serving HoE on 2× A100 80G GPUs. All experts share the same backbone network, without memory replication or cache isolation, making HoE significantly faster in practice.
>
> ---
>
> We hope this clarification resolves the misunderstanding. HoE is explicitly designed to reduce—not significantly increase—the parameter and deployment cost of multi-objective alignment.

---

> ### Author Response · Authors · 2025-11-21
> **Rebuttal - Part IV**
>
> > ###  Are there other ways to eliminate the need to train the proposed model besides HoE?
>
> Yes. As summarized in Table 1, several recent decoding-based methods—such as **Args , PAD**(2024), **GenARM, and RARM** (2025)—can also avoid additional training because they support mixed decoding over multiple pretrained models, even across different architectures.
> We fully acknowledge that these decoding-based approaches offer a certain advantage in this regard:
> **they can combine heterogeneous models without requiring any parameter updates.**
> However, their decoding paradigm inherently **imposes 2–3× inference latency**, lacks support from efficient inference engines, and suffers from comparative lower alignment performance in our experiments.
> These limitations make pure decoding-based approaches difficult to deploy in real multi-objective alignment systems. HoE, by contrast, achieves training efficiency without sacrificing inference speed or performance, and remains compatible with existing high-throughput inference stacks.
>
> > ### Are there any additional baselines that were published in this year to be taken as baselines in the experiments?
>
> Thank you for this helpful suggestion. We have added two newly published 2025 test-time alignment baselines:
> - **PARM** — Multi-Objective Test-Time Alignment via Preference-Aware Autoregressive Reward Model (*ICML 2025*)
> - **GenARM** — Reward-Guided Generation with Autoregressive Reward Model for Test-Time Alignment (*ICLR 2025*)
>
> We have included full comparisons in **Appendix Table 6**, along with representative performance tables immediately below. These updated results further validate the competitiveness of HoE against the latest state-of-the-art baselines.
>
>
> Setting	| [1,0,0]	| [0.5,0.5,0]	| [0,1,0]	| [0,0.5,0.5]	| [0,0,1]	| [0.5,0,0.5]	| [0.33,0.33,0.33]
> ------ | ------ | ------ | ------ | ------ | ------ | ------ | ------
> PAD             | 1.41 | **1.25** | 1.12 | 0.93  | 0.86  | 1.08 | **1.06** |
> GenARM          | 1.75 | 0.99 | 1.53 | 1.08  | 1.46  | 0.86 | 0.73 |
> PARM            | 2.05 | 1.21 | 1.87 | 1.34  | 1.46  | 1.03 | 0.86 |
> HoE		 | **3.05** | 1.02 | **2.25** | **1.79**  | **2.03**  | **1.78** | 1.03 |
>
>
> Setting	| [1,0,0]	| [0.5,0.5,0]	| [0,1,0]	| [0,0.5,0.5]	| [0,0,1]	| [0.5,0,0.5]	| [0.33,0.33,0.33]
> ------ | ------ | ------ | ------ | ------ | ------ | ------ | ------
> PAD             | 0.94 | 0.95 | 1.05 | 1.02  | 1.10  | 0.93 | 0.96 |
> GenARM          | 1.27 | 0.84 | 1.08 | 0.65  | 0.64  | 0.76 | 0.53 |
> PARM            | 1.38 | **0.95** | 1.15 | 0.76  | 0.78  | 0.88 | 0.72 |
> HoE			 | **2.36**  | 0.66 | **1.5** | **1.58** | **1.96**  | **1.63**  | **1.01** |
>
> ---
> ---
> ---
>
> ### Once again, we sincerely appreciate the reviewer’s time and thoughtful feedback. Your comments have helped us significantly strengthen the clarity and completeness of our manuscript. If any further concerns remain, we would be more than happy to address them.

---

### Author Response · Authors · 2025-12-02
**Summary of our efforts during the rebuttal stage**

We sincerely thank the reviewers and the AC for their time and the insightful feedback. During the rebuttal stage, we made substantial clarifications, additions, and corrections that significantly strengthened the paper. Below is a concise summary of our efforts.

---

## 1. Clarifying the Motivation and Necessity of HoE’s Design

Several reviewers misunderstood why HoE combines Multi-Objective LoRA Experts and Router Experts. We provided a detailed clarification from both empirical and conceptual perspectives.

**Conceptually**, HoE is driven by two core principles:
(1) Decomposing multi-objective alignment into local experts, and
(2) Employing a hierarchical expert structure to balance performance and parameter efficiency.

**Empirically**, we conducted extensive ablations in 2-objective (Figure.5 Left), 3-objective (Appendix Table.4) , and 5-objective settings (Appendix Table.5) . These experiments consistently show that:
- Router experts reliably improve PF coverage,
- Their benefits cannot be replicated by LoRA expansion alone, and
- LoRA experts and router experts complement each other,  providing the best performance–cost trade-off only when combined.

We further highlighted the strengths of router experts:
- **Negligible parameter overhead** (verified via cost analysis in Table3),
- **Performance comparable to adding an extra LoRA expert** (extensive ablations study), and
- **Fine-grained, dynamic, token-level routing** (verified via case study in Fig. 6).

To address concerns on motivation, we also provided training-process comparisons (anonymous link), demonstrating the advantages of decomposition in efficiency, modularity, stability, and avoiding gradient conflict.

---

## 2. Addressing Misunderstandings about Overhead

Reviewers expressed concerns about model size (cfcq) and hyperparameter overhead (V6P4). We clarified these with quantitative evidence:

- In Appendix B.2, we showed that HoE adds minimal parameter overhead and incurs no meaningful inference-latency penalty
- We listed all hyperparameters used in merging and compression, showing that only 2–3 hyperparameters require tuning and the total tuning time (<1 hour) is negligible

Thus, HoE remains **lightweight, plug-and-play**, and consistent with the “training-free merging” paradigm widely accepted in the literature.

---

## 3. Supplementing Missing Critical Details

In response to Reviewers V6P4 and tavC, we added complete missing details that are essential for reproducibility:
- All hyperparameters and their search ranges,
- The source and construction of LoRA parameters,
- The exact Merge function (PCB-merging + DARE),
- The ASVD-based compression pipeline and its alternatives.

---

## 4. Adding Newly Published Baselines (2025)

Per reviewer request, we added strong 2025 baselines: GenARM  and PARM
Comparisons are included in **Appendix Table 6**, demonstrating that HoE maintains competitive or superior performance.


---

## 5. Fixing Writing Issues and Mathematical Inconsistencies

We corrected typos, inaccurate expressions, ambiguous terminology and the inconsistent mathematical formulations

---

## 6. Addressing Evaluation Concerns
In response to , we clarified that our evaluation integrates:
- Reward-model scores,
- Benchmark-based evaluation,
- Rule-based verification,
- GPT-4 win-rate assessments.

This suite covers all evaluation paradigms used in prior multi-objective alignment work.

---

We genuinely appreciate the reviewers’ constructive comments. We hope that the strengthened manuscript now fully demonstrates the merit, novelty, and significance of HoE for multi-objective alignment in LLMs.

---

### Meta-Review · Area_Chair_kERK · 2025-12-27

**Summary:**

Most reviewers (**V6P4, tavC**) agree that HoE (Hierarchical Mixture-of-Experts) is a novel approach to multi-objective alignment. All reviewers (**cfcq, V6P4, tavC**) also find the ablation studies comprehensive and strong.

At the same time, all three reviewers raise concerns about the motivation and potential over-complication of the design. In particular, **V6P4** and **tavC** argue that the multi-objective router expert may be redundant, while **cfcq** worries about the additional parameter overhead introduced by combining LoRA and router experts. **V6P4** and **tavC** also flag missing or unclear technical details—especially around model merging, hyperparameter selection, and overall reproducibility. **V6P4** further questions parts of the convergence analysis in Appendix G. Finally, **tavC** notes that relying heavily on reward-model evaluation may not fully reflect objective, user-centric response quality.

In the rebuttal, the authors strengthen the motivation for HoE with additional experiments (Appendix Tables 4 and 5), arguing that multi-objective LoRA experts and router experts play complementary roles and are both necessary for strong performance. They also provide quantitative evidence that HoE introduces minimal parameter overhead and no meaningful inference-latency penalty (Appendix B.2). The revised manuscript further adds missing implementation details, including the merging procedure, the hyperparameter search space, and clarifications on reward-model selection. However, the rebuttal does not fully resolve the concerns about the convergence-rate proof in Appendix G raised by **V6P4**.

Overall, given the novelty of the approach, the strength of the empirical results, and the fact that most reviewer concerns have been addressed, I recommend acceptance. The work has the potential to make a positive impact on parameter-efficient multi-objective alignment.

**Reviewer Concerns:**

Three reviewers share the concern on the motivation of HoE, especially around why it needs both the multi-objective LoRA experts and router experts. This is addressed in the rebuttal. The author conducted additional experiments on  2-objective, 3-objective, and 5-objective settings with different number of LoRA experts and router experts. The new experiments show that the LoRA experts and router experts are complimentary to each other and both contributes to the final performance in the pareto-frontier.

Reviewer **cfcq** has concern on size of additional parameters in HoE. The authors address the concern by showing the storage, training parameters & costs, inference cost in Appendix B.2.

Reviewer **V6P4** and **tavC** have concern on the missing details about the merge function, hyperparameter selection and reproducibility. The authors add more details about the merging function  (PCB-merging + DARE) in the updated manuscript, hyperparameter search space and the HPO method.

Reviewer **tavC** has concern on the reward model selection. This is addressed by the rebuttal, where author mentions the rationale for the model-based RMs, which have been carefully selected for each scoring aspect and are scalable. The author also cross-validate the performance with non-RM-based metrics for Math & Code datasets.

Reviewer **V6P4**’s concern on the convergence rate proof is not adequately addressed. The author did not respond to the concern around “*Second, even if local stationary point convergence is established, a local stationary point is not, by any means, equivalent to a Pareto optimal solution.*”

**Reviewer Scores:**

I think both reviewer cfcq and V6P4 may keep or increase their score since the most critical concerns around motivation and technical details are addressed. Reviewer tavC may increase the score (potentially, 4 —> 6) since all three concerns on methodology, reproducibility and evaluation are addressed by the rebuttal.

---

### Decision · Program_Chairs · 2026-01-26

Accept (Poster)